# HierarchicalContrast: A Coarse-to-Fine Contrastive Learning Framework for Cross-Domain Zero-Shot Slot Filling

**Junwen Zhang** and **Yin Zhang** *

College of Computer Science and Technology

Zhejiang University

{junwenzhang, yinzh}@zju.edu.cn

## Abstract

In task-oriented dialogue scenarios, cross-domain zero-shot slot filling plays a vital role in leveraging source domain knowledge to learn a model with high generalization ability in unknown target domain where annotated data is unavailable. However, the existing state-of-the-art zero-shot slot filling methods have limited generalization ability in target domain, they only show effective knowledge transfer on seen slots and perform poorly on unseen slots. To alleviate this issue, we present a novel **Hi**erarchical **C**ontrastive **L**earning Framework (**HiCL**) for zero-shot slot filling. Specifically, we propose a *coarse- to fine-grained* contrastive learning based on Gaussian-distributed embedding to learn the generalized deep semantic relations between utterance-tokens, by optimizing inter- and intra-token distribution distance. This encourages HiCL to generalize to slot types unseen at training phase. Furthermore, we present a new *iterative label set semantics inference* method to unbiasedly and separately evaluate the performance of unseen slot types which entangled with their counterparts (i.e., seen slot types) in the previous zero-shot slot filling evaluation methods. The extensive empirical experiments[1] on four datasets demonstrate that the proposed method achieves comparable or even better performance than the current state-of-the-art zero-shot slot filling approaches.

## 1 Introduction

Slot filling models are devoted to extracting the contiguous spans of tokens belonging to pre-defined slot types for given spoken utterances and gathering information required by user intent detection, and thereby are an imperative module of task-oriented dialogue (TOD) systems. For instance, as shown in

---

*Corresponding author

[1]The offcial implementation of HiCL is available at https://github.com/ai-agi/HiCL.

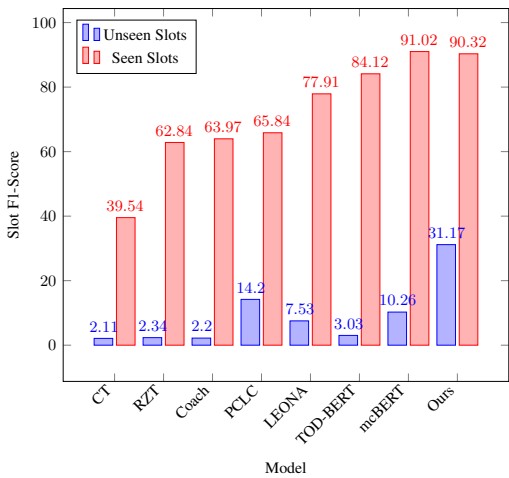

Figure 1: Cross-domain slot filling models' performance on unseen and seen slots in `GetWeather` target domain on SNIPS dataset.

Figure 2, given a user utterance "send a reminder for a tire check next week" belonging to reminder domain, the slot filling task is to identify **slot entities**: "a tire check" and "next week" that correspond to **slot types** (an alias of ***slot type*** is ***slot***), *todo* and *date_time*, respectively.

Supervised slot filling methods (Kurata et al., 2016; Wang et al., 2018; Li et al., 2018; Goo et al., 2018; Qin et al., 2019) have achieved promising performance. Nevertheless, these methods are strongly dependent on substantial and high-quality annotation data for each slot type, which prevents them from transferring to new domains with little or no labeled training samples.

To solve this problem, more approaches have emerged to deal with this data scarcity issue by leveraging zero-shot slot filling (ZSSF). Typically, these approaches can be divided into two categories: one-stage and two-stage. In one-stage paradigm, Bapna et al. (2017); Shah et al. (2019); Lee and Jha (2019) firstly generate utterance representation in token level to interact with the representation of each slot description in semantic space,

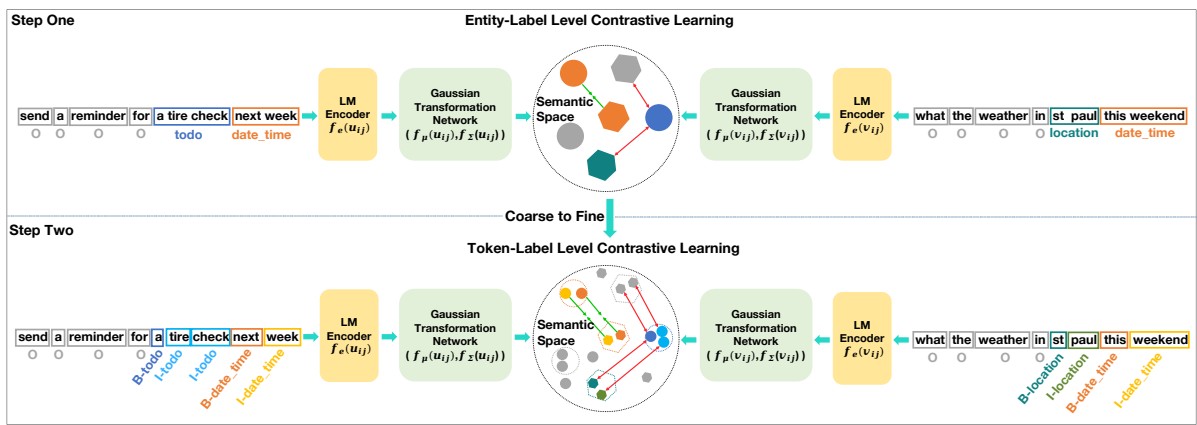

Figure 2: Hierarchical contrastive learning (CL), where coarse-grained and fine-grained slot labels are used as supervised signal for CL, respectively, i.e., step one is entity-label level CL and step two is token-label level CL. Entity-label is a pseudo-label in our Hierarchical CL. Different colors of rectangular bounding box denote different slot types.

and then predict each slot type for utterance token individually. The primary weakness for this paradigm is multiple prediction issue where a token will probably be predicted as multiple slot types (Liu et al., 2020; He et al., 2020). To overcome this issue, Liu et al. (2020); He et al. (2020); Wang et al. (2021) separate slot filling task into two-stage paradigm. They first identify whether the tokens in utterance belong to BIO (Ramshaw and Marcus, 1995) entity span or not by a binary classifier, subsequently predict their specific slot types by projecting the representations of both slot entity and slot description into the semantic space and interact on each other. Based on the previous works, Siddique et al. (2021) propose a two-stage variant, which introduces linguistic knowledge and pretrained context embedding, along with the entity span identify stage (Liu et al., 2020), to promote the effect on semantic similarity modeling between slot entity and slot description. Recently, Heo et al. (2022) develop another two-stage variant that applies momentum contrastive learning technique to train BERT (Devlin et al., 2019a) and to improve the robustness of ZSSF. However, as shown in Figure 1, we found that these methods perform poorly on unseen slots in the target domain.

Although two-stage approaches have flourished, one-pass prediction mechanism of these approaches (Liu et al., 2020; He et al., 2020; Wang et al., 2021) inherently limit their ability to infer unseen slots and seen slots separately. Thus they have to adopt the biased test set split method of unseen slots (see more details in Appendix F), being incapable of faithfully evaluating the real unseen slots

performance. Subsequently, their variants (Siddique et al., 2021; Heo et al., 2022; Luo and Liu, 2023) still struggle in the actual unseen slots performance evaluation due to following this biased test set split of unseen slots (Siddique et al., 2021; Heo et al., 2022), or the intrinsic architecture limit of one-pass inference (Luo and Liu, 2023). In another line (Du et al., 2021; Yu et al., 2021), ZSSF is formulated as a one-stage question answering task, but it is heavily reliant upon handcrafted question templates and ontologies customized by human experts, which is prohibitively expensive for this method to generalize to unseen domains. Besides, the problem with multiple slot types prediction that happened to them (Siddique et al., 2021; Heo et al., 2022; Du et al., 2021; Yu et al., 2021) seriously degrades their performance. In this paper, we introduce a new *iterative label set semantics inference* method to address these two problems.

Moreover, a downside for these two orthogonal methods is that they are not good at learning *domain-invariant* features (e.g., generalized token-class properties) by making full use of sourcedomain data while keeping target-domain of interest unseen. This may lead them to overfit to the limited slot types in source training domains. Actually the current models' performance on unseen slots in target domain is still far from upper bound.

To tackle the above limitation, intuitively, through contrastive learning (CL), we can redistribute the distance of token embeddings in semantic space to learn generalized token-class features across tokens, and better differentiate between token classes, and even between new classes (*slot-*

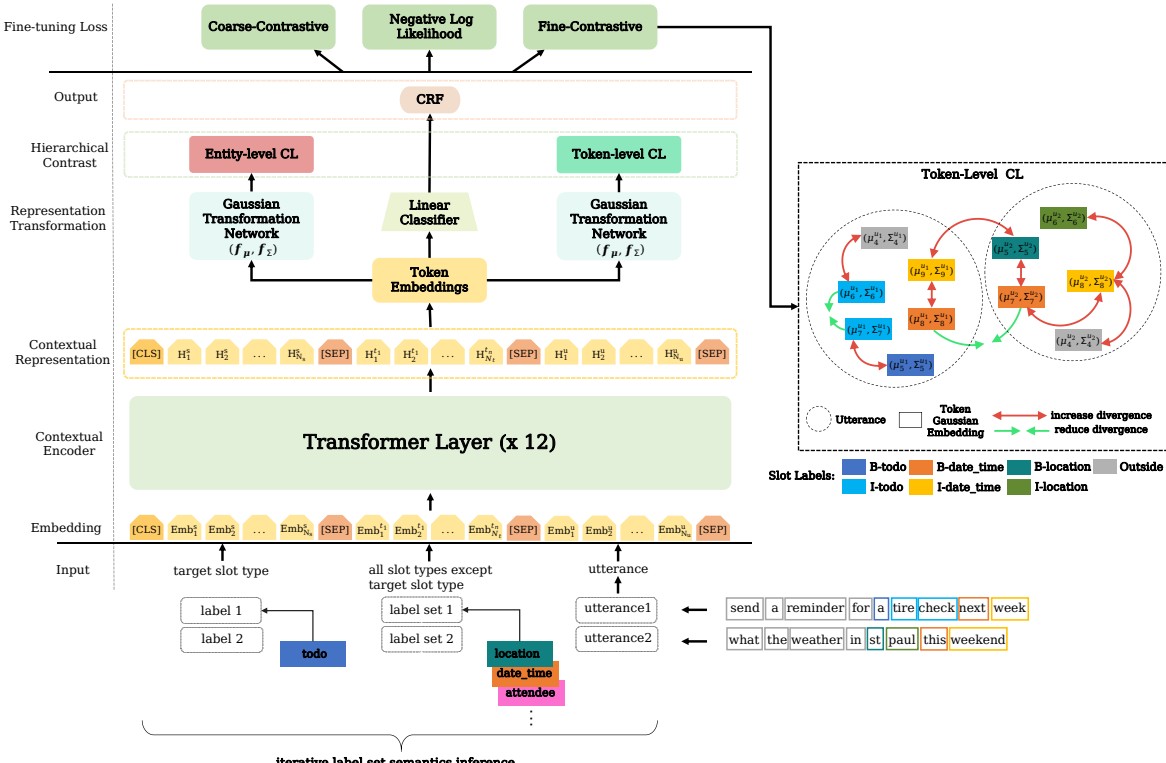

Figure 3: The main architecture of HiCL. For simplicity, we only draw detailed illustration for fine-grained token-level CL. Different colors of rectangular box in utterances and token-level CL (right side) denote different slot types.

*agnostic* features) , which is beneficial for new token class generalization across domains (Das et al., 2022). However, it's hard to train token-level class since its supervised labels are closely distributed, which easily leads to misclassifications (Ji et al., 2022). While training entity-level class is relatively easy since its training labels are dispersedly distributed and it does not require label dependency that exists in token-level sequence labeling training (Ji et al., 2022).

We argue that entity-level class knowledge contributes to token-level class learning, and their combination is beneficial for ZSSF task. Entity-level class learning supplements token-class learning with entity type knowledge and boundary information between entity and non-entity, which lowers the difficulty of token class training.

Hence, we propose a new hierarchical CL framework called **HiCL**, a coarse-to-fine CL approach. As depicted in Figure 2, it first **coarsely** learns entity-class knowledge of entity type and boundary knowledge via entity-level CL. Then, it combines features of entity type and boundary, and **finely** learns token-class knowledge via token-level CL.

In recent years, some researchers have employed

Gaussian embedding to learn the representations of tokens (Vilnis and McCallum, 2015; Mukherjee and Hospedales, 2016; Jiang et al., 2019a; Yüksel et al., 2021) due to their superiority in capturing the uncertainty in representations. This motivates us to employ Gaussian embedding in our HiCL to represent utterance-tokens more robustly, where each token becomes a density rather than a single point in latent feature space. Different from existing slot filling contrastive learners (He et al., 2020; Wu et al., 2020; Wang et al., 2021; Heo et al., 2022) that optimize training objective of pairwise similarities between point embeddings, HiCL aims to optimize distributional divergence by leveraging effectively modeling Gaussian embeddings. While point embeddings only optimize pairwise distance, Gaussian embeddings also comprise additional constraint which preserves the class distribution through their variance estimates. This distinctive quality helps to explicitly model entity- or token- class distributions, which not only encourages HiCL to learn generalized feature representations to categorize and differentiate between different entity (token) classes, but also fosters zero-sample target domain adaptation.

Concretely, as shown in Figure 3, our token-level CL pulls inter- and intra-token distributional distance with similar labels closer in semantic space, while pushing apart dissimilar ones. Gaussian distributed embedding enables token-level CL to better capture semantics uncertainty and semantics coverage variation of token-class than point embedding. This facilitates HiCL to better model generalized *slot-agnostic* features in cross-domain ZSSF scenario.

Our major contributions are three-fold:

- We introduce a novel *hierarchical CL* (coarse-to-fine CL) approach based on Gaussian embedding to learn and extract *slot-agnostic* features across utterance-tokens, effectively enhancing the model's generalization ability to identify new slot-entities.

- We find unseen slots and seen slots *overlapping* problem in the previous methods for unseen slots performance evaluation, and rectify this bias by splitting test set from slot type granularity instead of sample granularity, thus propose a new *iterative label set semantics inference* method to train and test unseen slots separately and unbiasedly. Moreover, this method is also designed to relieve the multiple slot types prediction issue.

- Experiments on two evaluation paradigms, four datasets and three backbones show that, compared with the current state-of-the-art (SOTA) models, our proposed HiCL framework achieves competitive unseen slots performance, and overall performance for cross-domain ZSSF task.

## 2 Problem Definition

**Zero-shot Setting** For ZSSF, a model is trained in source domains with a slot type set $\{\mathcal{A}^s_{(i)}\}$ and tested in new target domain with a slot type set $\{\mathcal{A}^t\} = \{\mathcal{A}^{ts}_{(j)}\} \cup \{\mathcal{A}^{tu}_{(k)}\}$ where $i$, $j$ and $k$ are index of different slot type sets, $\mathcal{A}^{ts}_{(j)}$ are the slot types that both exist in source domains and target domain (seen slots), and $\mathcal{A}^{tu}_{(k)}$ are the slot types that only exist in target domain (unseen slots). Since $\{\mathcal{A}^s_{(i)}\} \cap \{\mathcal{A}^{tu}_{(k)}\} = \emptyset$, it is a big challenge for the model to generalize to unseen slots in target domain.

**Task Formulation** Given an utterance $\mathcal{U} = \{x_1, ..., x_n\}$, the task of ZSSF aims to output a label sequence $O = \{o_1, ..., o_n\}$, where $n$ is the length of $\mathcal{U}$.

## 3 Methodology

The architecture of HiCL is illustrated in Figure 3. HiCL adopts an *iterative label set semantics inference* enhanced *hierarchical CL* approach and conditional random field (CRF) scheme. Firstly, HiCL successively employs entity-level CL and token-level CL, to optimize distributional divergence between different entity- (token-) class embeddings. Then, HiCL leverages generalized entity (token) representations refined in the previous CL steps, along with the slot-specific features learned in the CRF training with BIO label, to better model the alignment between utterance-tokens and slot-entities.

### 3.1 Hierarchical Contrastive Learning

**Encoder** Given an utterance of $n$ tokens $\mathcal{U} = \{x_i\}^n_{i=1}$, a target slot type with $k$ tokens $\mathcal{S} = \{s_i\}^k_1$, and all slot types except the target slot type with $q$ tokens $\mathcal{A} = \{a_i\}^q_1$, we adopt a pre-trained BERT (Devlin et al., 2019a) as the encoder and feed "$[CLS]\mathcal{S}[SEP]\mathcal{A}[SEP]\mathcal{U}[SEP]$" as input to the encoder to obtain a $d_l$-dimensional hidden representation $\boldsymbol{h}_i \in \mathbb{R}^{d_l}$ of each input instance:

$$\mathcal{H} = \mathrm{BERT}([CLS]\mathcal{S}[SEP]\mathcal{A}[SEP]\mathcal{U}[SEP]) \tag{1}$$

where $\mathcal{H} = \{\boldsymbol{h}_i\}^m_{i=1}$, $\mathcal{H} \in \mathbb{R}^{m \times d_l}$, $\mathcal{S} \cap \mathcal{A} = \emptyset$. We adopt two encoders for our HiCL, i.e., BiLSTM (Hochreiter and Schmidhuber, 1997) and BERT(Devlin et al., 2019a), to align with baselines.

**Gaussian Transformation Network** We hypothesize that the semantic representations of encoded utterance-tokens follow Gaussian distributions. Similar to (Jiang et al., 2019a; Das et al., 2022), we adopt exponential linear unit (ELU) and non-linear activation functions $f_\mu$ and $f_\Sigma$ to generate Gaussian mean and variance embedding, respectively:

$$\begin{aligned} \boldsymbol{\mu}_i &= f_u(\boldsymbol{h}_i) \\ \boldsymbol{\Sigma}_i &= \mathrm{ELU}(f_\Sigma(\boldsymbol{h}_i)) + \mathbf{1} \end{aligned} \tag{2}$$

where $\mathbf{1} \in \mathbb{R}^d$ is an array with all value set to 1, ELU and $\mathbf{1}$ are designed to ensure that every element of variance embedding is non-negative. $\boldsymbol{\mu}_i \in \mathbb{R}^d$, $\boldsymbol{\Sigma}_i \in \mathbb{R}^{d \times d}$ denotes mean and diagonal

covariance matrix of Gaussian embeddings, respectively. $f_\mu$ and $f_\Sigma$ are both implemented with ReLU and followed by one layer of linear transformation.

**Positive and Negative Samples Construction** In the training batch, given the anchor token $x_u$, if token $x_v$ shares the same slot label with token $x_u$, i.e., $y_v = y_u$, then $x_v$ is the positive example of $x_u$. Otherwise, if $y_v \neq y_u$, $x_v$ is the negative example of $x_u$.

**Contrastive Loss** Given a pair of positive samples $x_u$ and $x_v$, their Gaussian embedding distributions follow $x_u \sim \mathcal{N}(\boldsymbol{\mu}_u, \boldsymbol{\Sigma}_u)$ and $x_v \sim \mathcal{N}(\boldsymbol{\mu}_v, \boldsymbol{\Sigma}_v)$, both with $m$ dimensional. Following the formulas (Iwamoto and Yukawa, 2020; Qian et al., 2021), the Kullback-Leibler divergence from $\mathcal{N}(\boldsymbol{\mu}_u, \boldsymbol{\Sigma}_u)$ to $\mathcal{N}(\boldsymbol{\mu}_v, \boldsymbol{\Sigma}_v)$ is calculated as below:

$$
\begin{aligned}
\mathcal{D}_{\mathrm{KL}}[\mathcal{N}_u||\mathcal{N}_v] &= \mathcal{D}_{\mathrm{KL}}\Big[\mathcal{N}(\boldsymbol{\mu}_u, \boldsymbol{\Sigma}_u)||\mathcal{N}(\boldsymbol{\mu}_v, \boldsymbol{\Sigma}_v)\Big] \\
&= \int \mathcal{N}(\boldsymbol{\mu}_u, \boldsymbol{\Sigma}_u)\mathcal{N}(\boldsymbol{\mu}_v, \boldsymbol{\Sigma}_v)dx \\
&= \frac{1}{2}\Big[(\boldsymbol{\mu}_u - \boldsymbol{\mu}_v)^T \boldsymbol{\Sigma}_v^{-1}(\boldsymbol{\mu}_u - \boldsymbol{\mu}_v) \\
&\quad + \log\frac{|\boldsymbol{\Sigma}_v|}{|\boldsymbol{\Sigma}_u|} - m \\
&\quad + \mathrm{T_r}(\boldsymbol{\Sigma}_v^{-1}\boldsymbol{\Sigma}_u)\Big]
\end{aligned}
\tag{3}
$$

where $\mathrm{T_r}$ is the trace operator. Since the asymmetry features of the Kullback-Leibler divergence, we follow the calculation method (Das et al., 2022), and calculate both directions and average them:

$$
s(u, v) = \frac{1}{2}\big(\mathcal{D}_{\mathrm{KL}}[\mathcal{N}_u||\mathcal{N}_v] + \mathcal{D}_{\mathrm{KL}}[\mathcal{N}_v||\mathcal{N}_u]\big) \tag{4}
$$

Suppose the training set in source domains is $\mathcal{T}_a$, at each training step, a randomly shuffled batch $\mathcal{T} \in \mathcal{T}_a$ has batch size of $N_t$, each sample $(x_j, y_j) \in \mathcal{T}$. For each anchor sample $x_u$, match all positive instances $\mathcal{T}_u \in \mathcal{T}$ for $x_u$ and repeat it for all anchor samples:

$$
\mathcal{T}_u = \big\{(x_v, y_v) \in \mathcal{T} \mid y_v = y_u, u \neq v\big\} \tag{5}
$$

Formulating the Gaussian embedding loss in each batch, similar to Chen et al. (2020), we calculate the *NT-Xent* loss:

$$
\mathcal{L}^u = -\sum_{j=1}^{n_u} \log \frac{\exp(-s(u, j)/\tau)}{\sum_{k=1}^{K} \mathbb{1}_{[k \neq u]}\exp(-s(u, k)/\tau)}/n_u
\tag{6}
$$

where $\mathbb{1}_{[k \neq u]} \in \{0, 1\}$ is an indicator function evaluating to 1 iff $k \neq u$, $\tau$ is a scalar temperature parameter and $n_u$ is the total number of positive instances in $\mathcal{T}_u$.

**Coarse-grained Entity-level Contrast** In coarse-grained CL, entity-level slot labels are used as CL supervised signals in training set $\mathcal{T}_a$. Coarse-grained CL optimizes distributional divergence between tokens Gaussian embeddings and models the entity class distribution. According to Eq.(3)-Eq.(6), we can obtain coarse-grained entity-level contrastive loss $\mathcal{L}_{coarse}^i$, and the in-batch coarse-grained CL loss is formulated:

$$
\mathcal{L}_{coarse} = \frac{1}{N_t}\sum_{i=1}^{N_t}\mathcal{L}_{coarse}^i \tag{7}
$$

**Fine-grained Token-level Contrast** In fine-grained CL, token-level slot labels are used as CL supervised signals in training set $\mathcal{T}_a$. As illustrated in Figure 3, fine-grained CL optimizes KL-divergence between tokens Gaussian embeddings and models the token class distribution. Similarly, the in-batch fine-grained CL loss is formulated:

$$
\mathcal{L}_{fine} = \frac{1}{N_t}\sum_{i=1}^{N_t}\mathcal{L}_{fine}^i \tag{8}
$$

### 3.2 Training Objective

The training objective $\mathcal{L}$ is the weighted sum of regularized loss functions.

**Slot Filling Loss**

$$
\mathcal{L}_s := -\sum_{j=1}^{n}\sum_{i=1}^{n_l} \hat{y}_j^i \log\big(y_j^i\big) \tag{9}
$$

where $\hat{y}_j^i$ is the gold slot label of $j$-th token and $n_l$ is the number of all slot labels.

**Overall Loss**

$$
\mathcal{L} = \alpha\mathcal{L}_s + \beta\mathcal{L}_{coarse} + \gamma\mathcal{L}_{fine} + \lambda\|\Theta\| \tag{10}
$$

where $\alpha$, $\beta$ and $\gamma$ are tunable hyper-parameters for each loss component, $\lambda$ denotes the coefficient of $L_2$-regularization, $\Theta$ represents all trainable parameters of the model.

## 4 Experiments

### 4.1 Dataset

We evaluate our approach on four datasets, namely SNIPS (Coucke et al., 2018), ATIS (Hemphill et al., 1990), MIT_corpus (Nie et al., 2021) and SGD (Rastogi et al., 2020).

| Training Setting | | Zero-shot | | | | | | | |
|---|---|---|---|---|---|---|---|---|---|
| Method ⇓ | Domain ⇒ | AP | BR | GW | PM | RB | SCW | FSE | AVG F1 |
| *RNN based* | CT† | 38.82 | 27.54 | 46.45 | 32.86 | 14.54 | 39.79 | 13.83 | 30.55 |
| | RZT† | 42.77 | 30.68 | 50.28 | 33.12 | 16.43 | 44.45 | 12.25 | 32.85 |
| | Coach† | 50.90 | 34.01 | 50.47 | 32.01 | 22.06 | 46.65 | 25.63 | 37.39 |
| | CZSL-Adv† | 53.89 | 34.06 | 52.24 | 34.59 | 31.53 | 50.61 | 30.05 | 40.99 |
| | PCLC† | 59.24 | 41.36 | 54.21 | 34.95 | 29.31 | 53.51 | 27.17 | 42.82 |
| | HiCL+BiLSTM (ours) | 53.16 | 39.97 | 55.78 | 35.13 | 27.16 | 54.07 | 26.51 | 41.68 |
| *ELMo based** | LEONA‡ | 50.23 | 46.58 | 62.91 | 40.49 | 22.67 | 45.86 | 28.15 | 42.41 |
| | HiCL+ELMo (ours) | 52.86 | 48.67 | 64.35 | 42.40 | 27.38 | 49.96 | 30.33 | 45.14 |
| *BERT based* | TOD-BERT | 47.26 | 44.91 | 64.30 | 29.36 | 25.02 | 62.85 | 44.11 | 45.40 |
| | mcBERT♮ | 54.28 | 55.28 | 75.60 | 35.16 | 31.88 | 70.73 | 43.77 | 52.39 |
| | RCSF | 68.70 | 63.49 | 65.36 | 53.51 | 36.51 | 69.22 | 33.54 | 55.76 |
| | HiCL+BERT (ours) | 54.35 | 61.06 | 77.91 | 43.65 | 36.97 | 73.22 | 44.47 | 55.95 |
| | w/o coarse CL (ours) | 52.44 | 57.43 | 75.02 | 44.21 | 36.82 | 72.06 | 44.84 | 54.69 |
| | w/o fine CL (ours) | 55.10 | 53.68 | 77.44 | 44.94 | 34.14 | 70.63 | 40.54 | 53.78 |

Table 1: Slot F1 scores on SNIPS dataset for different target domains that are unseen in training. † denotes the results reported in (Wang et al., 2021). ‡ denotes that we run the publicly released code (Siddique et al., 2021) to obtain the experimental results and ♮ denotes that we reimplemented the model. AP, BR, GW, PM, RB, SCW and FSE denote AddToPlaylist, BookRestaurant, GetWeather, PlayMusic, RateBook, SearchCreativeWork and FindScreeningEvent, respectively. AVG denotes average.

| Training Setting | | Zero-shot | | | | | | |
|---|---|---|---|---|---|---|---|---|
| Method ⇓ | Domain ⇒ | AR | AF | AL | FT | GS | OS | AVG F1 |
| *ELMo based** | LEONA‡ | 51.36 | 96.56 | 93.11 | 84.95 | 50.93 | 84.70 | 76.93 |
| | HiCL+ELMo (ours) | 50.64 | 98.34 | 93.24 | 86.19 | 52.47 | 84.63 | 77.58 |
| *BERT based* | TOD-BERT | 44.67 | 96.54 | 89.02 | 85.50 | 60.97 | 75.80 | 75.42 |
| | mcBERT♮ | 63.55 | 98.56 | 92.82 | 88.10 | 74.42 | 81.35 | 83.13 |
| | HiCL+BERT (ours) | 69.54 | 98.09 | 92.20 | 85.96 | 78.17 | 82.41 | 84.40 |
| | w/o coarse CL (ours) | 65.23 | 98.75 | 94.20 | 87.90 | 69.96 | 71.08 | 81.19 |
| | w/o fine CL (ours) | 62.83 | 98.40 | 93.51 | 87.91 | 81.04 | 84.00 | 84.62 |

Table 2: Slot F1 scores on ATIS dataset for different target domains that are unseen in training. AR, AF, AL, FT, GS and OS denote Abbreviation, Airfare, Airline, Flight, Ground Service, Others, respectively.

| Training Setting | | Zero-shot | | |
|---|---|---|---|---|
| Method ⇓ | Domain ⇒ | Movie | Restaurant | AVG F1 |
| *BERT based* | TOD-BERT | 71.72 | 48.23 | 59.90 |
| | mcBERT♮ | 76.51 | 57.37 | 66.94 |
| | HiCL+BERT (ours) | **77.75** | **58.35** | **68.05** |
| | w/o coarse CL (ours) | 74.99 | 51.23 | 63.11 |
| | w/o fine CL (ours) | 75.15 | 57.63 | 66.39 |

Table 3: Slot F1 scores on MIT_corpus dataset for different target domains that are unseen in training.

| Training Setting | | Zero-shot | | | | |
|---|---|---|---|---|---|---|
| Method ⇓ | Domain ⇒ | Buses | Events | Homes | Rental Cars | AVG F1 |
| *BERT based* | TOD-BERT | **35.04** | 56.43 | 79.92 | 53.40 | 56.20 |
| | mcBERT♮ | 27.12 | 51.38 | 80.14 | **59.43** | 54.52 |
| | HiCL+BERT (ours) | 27.63 | 50.20 | 81.73 | 59.24 | 54.70 |
| | HiCL+TOD-BERT (ours) | 28.07 | **58.29** | **84.51** | 55.53 | **56.60** |
| | w/o coarse CL (ours) | 24.94 | 53.62 | 83.45 | 53.17 | 53.80 |
| | w/o fine CL (ours) | 29.36 | 45.99 | 77.81 | 56.39 | 52.39 |

Table 4: Slot F1 scores on SGD dataset for different target domains that are unseen in training. BS, ET, HE, RC denote Buses, Events, Homes, Rental Cars, respectively.

MIT_corpus (Nie et al., 2021).

### 4.4 Baselines

We compare the performance of our HiCL with the previous best models, the details of baseline models are provided in Appendix E.

### 4.5 Training Approach

**Training Sample Construction** The output of HiCL is a BIO prediction for each slot type. The training samples are of the pattern $(\mathcal{S}_t, \mathcal{A}_t, \mathcal{U}_i, \mathcal{Y}_i')$, where $\mathcal{S}_t$ represents a target slot type, $\mathcal{A}_t$ represents all slot types except $\mathcal{S}_t$, $\mathcal{U}_i$ represents an utterance, $\mathcal{Y}_i'$ represents BIO label for $\mathcal{S}_t$, all slot types $\mathcal{A} = \mathcal{S}_t \cup \mathcal{A}_t$, for simple depiction, hierarchical CL labels are omitted here. For a sample from given dataset with the pattern $(\mathcal{U}_i, \mathcal{Y}_i)$ that contains entities for $k$ slot types, $k$ positive training samples for $\mathcal{U}_i$ can be generated by setting each of $k$ slot types as $\mathcal{S}_t$ in turn and generating the corresponding $\mathcal{A}_t$ and $\mathcal{Y}_i'$. Then $m$ negative training samples for $\mathcal{U}_i$ can be generated by choosing slot types that belongs to $\mathcal{A}$ and does not appear in $\mathcal{U}_i$. For example, in Figure 3, the utterance "what the weather in st paul this weekend" has the original label "O O O O B-location I-location B-date_time I-data_time". The positive samples are formatted as ["location", ... , ... , "O O O O B I O O"] and ["date_time", ... , ... , "O O O O O O B I"]. While the negative samples are formatted as ["todo", ... , ... , "O O O O O O O O"] and ["attendee", ... , ... , "O O O O O O O O"].

**Iterative Label Set Semantics Inference** Itera-

### 4.2 Unseen and Seen Slots Overlapping Problem in Test Set

The problem description, and the proposed rectified method for unseen and seen slots test set split are presented in Appendix F.

### 4.3 Evaluation Paradigm

**Training on Multiple Source Domains and Testing on Single Target Domain** A model is trained on all domains except a single target domain. For instance, the model is trained on all domains of SNIPS dataset except a target domain GetWeather which is used for zero-shot slot filling capability test. This multiple training domains towards single target domain paradigm is evaluated on datasets SNIPS (Coucke et al., 2018), ATIS (Hemphill et al., 1990) and SGD (Rastogi et al., 2020).

**Training on Single Source Domain and Testing on Single Target Domain** A model is trained on single source domain and test on a single target domain. This single training domain towards single target domain paradigm is evaluated on dataset of

tively feeding the training samples constructed in § 4.5 into HiCL, the model would output BIO label for each target slot type. We named this training or predict paradigm *iterative label set semantics inference* (ILSSI). Algorithm 1 and 2 in Appendix elaborate on more details of ILSSI.

## 5 Experimental Results

### 5.1 Main Results

We examine the effectiveness of HiCL by comparing it with the competing baselines. The results of the average performance across different target domains on dataset of SNIPS, ATIS, MIT_corpus and SGD are reported in Table 1, 2, 3, 4, respectively, which show that the proposed method consistently outperforms the previous BERT-based and ELMo-based SOTA methods, and performs comparably to the previous RNN-based SOTA methods. The detailed results of seen-slots and unseen-slots performance across different target domains on dataset of SNIPS, ATIS, MIT_corpus and SGD are reported in Table 6, 7, 8, 9, respectively. On seen-slots side, the proposed method performs comparably to prior SOTA methods, and on unseen-slots side, the proposed method consistently outperforms other SOTA methods.

### 5.2 Quantitative Analysis

**Ablation Study** To study the contribution of different component of hierarchical CL, we conduct ablation experiments and display the results in Table 1 to Table 9.

The results indicate that, on the whole, both coarse-grained entity-level CL and fine-grained token-level CL contribute substantially to the performance of the proposed HiCL on different dataset. Specifically, taking the performance of HiCL on SNIPS dataset for example, as shown in Table 1, the removal of token-level CL ("w/o fine CL") sharply degrades average performance of HiCL by 2.17%, while the removal of entity-level CL ("w/o coarse CL") significantly drops average performance of HiCL by 1.26%. Besides, as shown in Table 6, removing entity-level CL ("w/o $\mathcal{L}_{coarse}$"), the unseen slots effect is drastically reduced by 4.61%, and removing token-level CL ("w/o $\mathcal{L}_{fine}$"), the unseen slots effect of the proposed model is considerably decreased by 4.01%.

**Coarse-grained CL vs. Fine-grained CL** On the basis of ablation study results (§ 5.2), our analyses are that, coarse-grained CL complements fine-

grained CL with entity-level boundary information and entity type knowledge, while fine-grained CL complements coarse-grained CL with token-level boundary information (BIO) and token class type knowledge. Their combination is superior to either of them and helps HiCL to obtain better performance.

**Unseen Slots vs. Seen Slots** As shown in Table 6, 7, 8, 9, the proposed HiCL significantly improves the performance on unseen slots against the SOTA models, while maintaining comparable seen slots performance, which verifies the effectiveness of our HiCL framework in cross-domain ZSSF task. From the remarkable improvements on unseen slot, we clearly know that, rather than only fitting seen slots in source domains, our model have learned generalized *slot-agnostic* features of entity-classes including unseen classes by leveraging the proposed hierarchical contrast method. This enables HiCL to effectively transfer *domain-invariant* knowledge of slot types in source domains to unknown target domain.

| model | seen | unseen |
|---|---|---|
| HiCL (BERT Backbone) | | |
| + Gaussian Embedding + KL-div. | **68.94** | **29.71** |
| + Point Embedding + Euclidean | 64.52 | 28.54 |
| + Point Embedding + Cosine | 68.30 | 26.86 |

Table 5: The ablation study of HiCL adopting different types of embedding on SNIPS dataset.

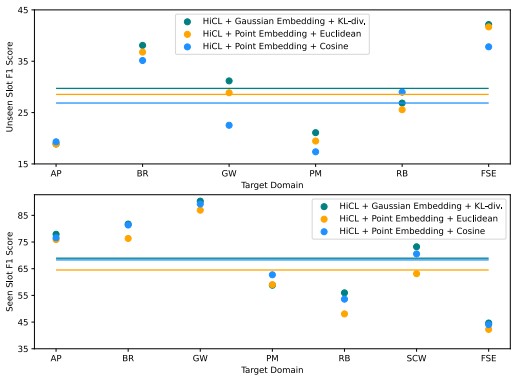

Figure 4: HiCL's performance on unseen-slots and seen slots of each domain in SNIPS dataset when equipped with Gaussian embedding and point embedding. The lines denote average F1 scores over all domains.

**Gaussian embedding vs. Point embedding** We provide ablation study for different forms of embedding that are migrated to our HiCL, i.e., Gaus-

sian embedding and point embedding, and investigate their performance impact on unseen slots and seen slots. As shown in Table 5 and Figure 4, HiCL achieves better performance on both seen slots and unseen slots by employing Gussian embedding over its counterpart, i.e., point embedding. This suggests that Gaussian embedding may be more suitable than point embedding for identifying slot entities of novel slot types in cross-domain generalization task.

**Multi-domains Training vs. Single-domain Training** From the results in Table 6, 7, 8, 9, we clearly see that, for unseen slots cross-domain transfer, single-domain training is much more difficult than multi-domain training. The averaged F1 score of unseen slots performance of HiCL across all target domains on three multi-domain training datasets (SNIPS, ATIS and SGD) is 38.97, whereas the averaged F1 score of unseen slots performance of HiCL on a single-domain training dataset of MIT_corpus is only 10.12. This tremendous gap may reveal that the diversity of source domains in training is a very critical factor that determines the model's capability of cross-domain migration of unseen slots. To have a closer analysis, plural source domains in training stage mean that there are abundant slot types that help the model learn generalized *domain-invariant* features of slot types, to avoid overfitting to the limited slot type classes in source domains. The results in Table 8 verify this analysis, we observe that, without the constraint of professionally designed generalization technique, learning with the limited slot types, TOD-BERT barely recognizes any unseen slot, and mcBERT performs poorly on unseen slots. Although HiCL achieves the best results against baselines due to specially designed generalization method of hierarchical CL, the proposed model also suffers from limited diversity of slot types in single-domain training mode, and its performance is significantly lower than that of multi-domain training.

**Performance Variation Analysis of TOD-BERT** Surprisingly, as shown in Table 1, 2, 3, TOD-BERT (current pre-trained TOD SOTA model) performs inferiorly on datasets of SNIPS, ATIS and MIT_corpus and fails to meet our expectations. We present analysis of causes as below: (1) TOD-BERT is unable to directly take advantage of the prior knowledge of pre-training on the datasets of SNIPS, ATIS and MIT_corpus. These datasets are not included in the pre-training corpora of TOD-

BERT and their knowledge remains unseen for TOD-BERT. (2) There is a discrepancy of data distribution between the corpora that TOD-BERT pre-trained on and the datasets of SNIPS, ATIS and MIT_corpus. The datasets that TOD-BERT pre-trained on are multi-turn task-oriented dialogues of modeling between user utterances and agent responses, whereas the datasets of SNIPS, ATIS and MIT_corpus are single-turn utterances of users in task-oriented dialogues. Perhaps this intrinsic data difference affects the performance of TOD-BERT on these single-turn dialogue datasets. (3) TOD-BERT may also suffer from catastrophic forgetting (Kirkpatrick et al., 2016) during pre-training. TOD-BERT is further pre-trained by initializing from BERT, catastrophic forgetting may prevent TOD-BERT from fully leveraging the general purpose knowledge of pre-trained BERT in zero-shot learning scenarios. From the experimental results, we observe that the performance of TOD-BERT is even much lower than BERT-based models (e.g., mcBERT), which may be a possible empirical evidence for the above analysis.

In contrast, TOD-BERT performs extremely well on SGD dataset and it beats all BERT-based models. This is because that TOD-BERT is pre-trained on SGD dataset (Wu et al., 2020) and it can thoroughly leverage the prior pre-trained knowledge on SGD to tackle various downstream tasks including zero-shot ones on this dataset. However, when our HiCL migrates to TOD-BERT (HiCL+TOD-BERT), as shown in Table 4 and 9, the performance of the model again achieves an uplift. Concretely, the overall performance increases by 0.4% and unseen slots performance increases by 9.18%, which is a prodigious boost, only at the expense of a drop of 2.43% on seen slots performance. This demonstrates that, in terms of unseen slots performance, even on the seen pre-training datasets, our method of HiCL can still compensate the shortcoming of pre-trained TOD models (e.g., TOD-BERT).

### 5.3 Qualitative Analysis

**Visualization Analysis** Figure 5 in Appendix provides t-SNE scatter plots to visualize the performance of the baseline models and HiCL on test set of GetWeather target domain of SNIPS dataset.

In Figure 5(a) and Figure 5(c), we observe that TOD-BERT and mcBERT have poor clustering styles, their representations of predicted slot en-

tities for unseen slots (digital number 1, 2, 3, and 4 in the figure) sparsely spread out in embedding space and intermingle with other slot entities' representations of unseen slots and `outside` (O) in large areas, and TOD-BERT is even more worse and its slot entities' representations of seen slots (digital number 5, 6, 7, 8 and 9 in the figure) are a little sparsely scattered. This phenomenon possibly suggests two problems. On the one hand, without effective constraints of generalization technique, restricted to prior knowledge of fitting the limited slot types in source training domains, TOD-BERT and mcBERT learn little *domain-invariant* and *slot-agnostic* features that could help them to recognize new slot-entities, they mistakenly classify many slot entities of unseen slots into `outside` (O). On the other hand, although TOD-BERT and mcBERT generate clear-cut clusters of slot-entity classes of seen slots, they possess sub-optimal discriminative capability between new slot-entity classes of unseen slots, they falsely predict multiple new slot types for the same entity tokens.

In Figure 5(b) and 5(d), we can see clearly that, HiCL produces a better clustering division between new slot-entity classes of unseen slots, and between new slot-entity class and `outside` (O), due to generalized differentiation ability between entity-class (token-class) by extracting class agnostic features through hierarchical CL. Moreover, equipped with Gaussian embedding and KL-divergence, HiCL exhibits even more robust performance on unseen slots than equipped with point embedding and Euclidean distance, the clusters of new slot-entity classes of unseen slots and `outside` (O) distribute more compactly and separately.

**Case Study**    Table 11 in Appendix demonstrates the prediction examples of ILSSI on SNIPS dataset with HiCL and mcBERT for both unseen slots and seen slots, in target domain of BookRestaurant and GetWeather, respectively. The main observations are summarized as follows:

(1) mcBERT is prone to repeatedly predict the same entity tokens for different unseen slot types, which leads to its misclassifications and performance degradation. For instance, in target domain of BookRestaurant, given the utterance "i d like a table for ten in 2 minutes at french horn sonning eye", mcBERT repeatedly predicts the same entity tokens "french horn sonning eye" for three different types of unseen slots. This phenomenon can be interpreted as a nearly random guess of slot type for certain entity, due to learning little prior knowledge of generalized token- or entity-classes, resulting in inferior capacity to differentiate between token- or entity-categories, which discloses the frangibility of mcBERT on unseen slots performance. Whereas, HiCL performs more robustly for entity tokens prediction versus different unseen slots, and significantly outperforms mcBERT on unseen slots in different target domains. Thanks to hierarchical CL and ILSSI, our HiCL learns generalized knowledge to differentiate between token- or entity-classes, even between their new classes, which is a generalized *slot-agnostic* ability. (2) HiCL is more capable of recognizing new entities over mcBERT by leveraging learned generalized knowledge of token- and entity-class. For instance, in target domain of GetWeather, both HiCL and mcBERT can recoginze token-level entity "warmer" and "hotter" that belong to unseen class of *condition_temperature*, but mcBERT fails to recognize "*freezing*" and "*temperate*" that also belong to the same class, owing to the limited generalization knowledge of token-level class. With the help of hierarchical CL that aims at extracting the most *domain-invariant* features of token- and entity-classes, our HiCL can succeed in recognizing these novel entities. (3) HiCL performs slightly better than mcBERT on the seen slots. The two models demonstrate equivalent knowledge transfer capability of seen slots from different training domains to target domains.

## 6    Conclusion

In this paper, we improve cross-domain ZSSF model from a new perspective: to strengthen the model's generalized differentiation ability between entity-class (token-class) by extracting the most domain-invariant and class-agnostic features. Specifically, we introduce a novel pretraining-free HiCL framework, that primarily comprises hierarchical CL and iterative label set semantics inference, which effectively improves the model's ability of discovering novel entities and discriminating between new slot-entity classes, which offers benefits to cross-domain transferability of unseen slots. Experimental results demonstrate that our HiCL is a backbone-independent framework, and compared with several SOTA methods, it performs comparably or better on unseen slots and overall performance in new target domains of ZSSF.

## Limitations and Future Work

Large language models (LLMs) exhibit powerful ability in zero-shot and few shot scenarios. However, LLMs such as ChatGPT seem not to be good at sequence labeling tasks (Li et al., 2023; Wang et al., 2023), for example, slot filling, named-entity recognition, etc. Our work endeavors to remedy this shortage with light-weighted language models. However, if the annotated datasets are large enough, our method will degenerate and even possibly hurt the generalization performance of the models (e.g., transformer based language models). Since the models would generalize pretty well through thoroughly learning the rich data features, distribution and dimensions, without the constraint of certain techniques that would become a downside under these circumstances, which reveals the principle that the upper bound of the model performance depends on the data itself. We directly adopt slot label itself in contrastive training of our HiCL, which does not model the interactions between label knowledge and semantics. In our future work, we will develop more advanced modules via label-semantics interactions, that leverages slot descriptions and pre-trained transformer-based large language models to further boost unseen slot filling performance for TOD.

## Acknowledgement

We thank Xiangfeng Li for grammar revision, Jinyan Wang for data statistics, Fanqi Shen for histogram plotting, and anonymous reviewers for their helpful suggestions. This work is supported by China Knowledge Centre for Engineering Sciences and Technology (CKCEST-2022-1-7).

## Ethics Statement

Our contribution in this work is fully methodological, namely a Gaussian embedding enhanced coarse-to-fine contrastive learning (HiCL) to boost the performance of cross-domain zero-shot slot filling, especially when annotated data is not available in new target domain. Hence, this contribution has no direct negative social impacts.

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

# Appendix

---

**Algorithm 1:** Iterative Label Set Semantics Inference

```
// training in source domain
```
**Input:** Batch Samples $\mathcal{D} = \sum_{i=1}^{n_{trb}} \mathcal{D}_i$,
where $\mathcal{D}_i = (\mathcal{S}_t, \mathcal{A}_t, \mathcal{U}_i, \mathcal{Y}_i', \mathcal{Y}_i^c, \mathcal{Y}_i^f)$
```
// § 4.5
// 𝒴ᵢᶜ and 𝒴ᵢᶠ denote coarse- and
   fine-grained CL label.
```
**Output:** Batch Loss $\mathcal{L}$
```
// Eq.10
```
**Data:** Training Data $\mathcal{D}_{tr}$

**1** **for** *training batch* $\mathcal{D} \in \mathcal{D}_{tr}$ **do**
**2**     **for** $\mathcal{D}_i \in \mathcal{D}$ **do**
**3**        $\mathcal{O}_i =$ CRF$(W_i \otimes (\text{PLM}((\mathcal{S}_t, \mathcal{A}_t, \mathcal{U}_i)) + b_i))$
**4**        $\mu_i = f_u(\text{PLM}((\mathcal{S}_t, \mathcal{A}_t, \mathcal{U}_i)))$ `// Eq.2`
**5**        $\Sigma_i = ELU(f_\Sigma(\text{PLM}((\mathcal{S}_t, \mathcal{A}_t, \mathcal{U}_i)))) + \mathbf{1}$
       `// Eq.2`
**6**     calculate $\mathcal{L}_{coarse}, \mathcal{L}_{fine}, \mathcal{L}_s$
**7**     calculate $\mathcal{L}$
**8**     back-propagate and update parameters to optimize $\mathcal{L}$

---

## A  Related Work

### A.1  Gaussian Embedding

Vilnis and McCallum (2015) initially explore to learn word embedding in Gaussian distribution space, they find that density-based Gaussian embedding helps capturing uncertainty of word representation and presenting more natural expression for asymmetries than point embedding. Wang et al. (2017) incorporate Gaussian distribution embedding into deep CNN architectures through an end-to-end pattern to discriminate first- and second-order image characteristics, which leverages the rich geometry and smooth representations of Gaussian embedding. Jiang et al. (2019b) employ Gaussian embedding in convolutional operations to capture the uncertainty of users preferences in recommendation system. Qian et al. (2021) advocate a contextualized Gaussian embedding that integrates inner-word knowledge and outer-word contexts into word representations and capture their more accurate semantics. Das et al. (2022) leverage Gaussian embedding in contrastive learning for few-shot named entity recognition task, which is a work closer to ours.

However, our work is fundamentally different from the research (Das et al., 2022) in many aspects. First of all, Das et al. (2022) present the method of entity level CL (entity-tokens level CL)

---

**Algorithm 2:** Iterative Label Set Semantics Inference

```
// prediction in target domain
```
**Input:** All Test Samples $\mathcal{D} = \sum_{i=1}^{n_{ts}} \mathcal{D}_i$,
, where $\mathcal{D}_i = (\mathcal{S}_t, \mathcal{A}_t, \mathcal{U}_i, \mathcal{Y}_i')$
**Output:** Total Prediction Set $\mathcal{O}$, $\mathcal{O}_{unseen}$ and $\mathcal{O}_{seen}$, and F1 Scores for $\mathcal{O}$, $\mathcal{O}_{unseen}$ and $\mathcal{O}_{seen}$ in Target Domain
```
// Testing Data 𝒟_ts includes all negative
   and positive samples constructed as § 4.5
   in Target Domain
```
**Data:** Testing Data $\mathcal{D}_{ts}$, All Slot Types in Source Training Domains $\mathcal{A}^{tr}$, All Slot Types in Target Domain $\mathcal{A}^{ts}$

**1** **for** *testing sample* $\mathcal{D}_i \in \mathcal{D}_{ts}$ **do**
**2**     **if** $\mathcal{S}_t \in \mathcal{A}^{ts} and \mathcal{S}_t \notin \mathcal{A}^{tr}$ **then**
**3**        $\mathcal{D}_{ts}^{unseen} \leftarrow \mathcal{D}_i$ `// add 𝒟ᵢ to 𝒟_ts^unseen`
**4**     **else if** $\mathcal{S}_t \in \mathcal{A}^{ts} and \mathcal{S}_t \in \mathcal{A}^{tr}$ **then**
**5**        $\mathcal{D}_{ts}^{seen} \leftarrow \mathcal{D}_i$ `// add 𝒟ᵢ to 𝒟_ts^seen`
**6** **for** *testing batch* $\mathcal{D} \in \mathcal{D}_{ts}$ **do**
**7**     **for** $\mathcal{D}_i \in \mathcal{D}$ **do**
**8**        $\mathcal{O}_i =$ CRF$(W_i \otimes (\text{PLM}((\mathcal{S}_t, \mathcal{A}_t, \mathcal{U}_i)) + b_i))$
**9**        $\mathcal{O} \leftarrow \mathcal{O}_i$ `// add 𝒪ᵢ to 𝒪`
**10** **for** *unseen slots testing batch* $\mathcal{D}_i^{un} \in \mathcal{D}_{unseen}$ **do**
**11**     **for** $\mathcal{D}_i \in \mathcal{D}_i^{un}$ **do**
**12**        $\mathcal{O}_i =$ CRF$(W_i \otimes (\text{PLM}((\mathcal{S}_t, \mathcal{A}_t, \mathcal{U}_i)) + b_i))$
**13**        $\mathcal{O}_{unseen} \leftarrow \mathcal{O}_i$ `// add 𝒪ᵢ to 𝒪_unseen`
**14** **for** *seen slots testing batch* $\mathcal{D}_i^{sn} \in \mathcal{D}_{seen}$ **do**
**15**     **for** $\mathcal{D}_i \in \mathcal{D}_i^{sn}$ **do**
**16**        $\mathcal{O}_i =$ CRF$(W_i \otimes (\text{PLM}((\mathcal{S}_t, \mathcal{A}_t, \mathcal{U}_i)) + b_i))$
**17**        $\mathcal{O}_{seen} \leftarrow \mathcal{O}_i$ `// add 𝒪ᵢ to 𝒪_seen`
**18** calculate f1 score for $\mathcal{O}$
**19** calculate f1 score for $\mathcal{O}_{unseen}$
**20** calculate f1 score for $\mathcal{O}_{seen}$

---

in NER task, while it is the first of its kind for us to introduce token level CL, and we innovatively present a hierarchical CL architecture and empirically verify that the combination of entity- and token- level CL will significantly outperform either of them. Secondly, we explore a more challenge research orientation, i.e., zero-shot task of single-turn and multi-turn task-oriented dialogues instead of few-shot NER task that comprises single independent sentences (Das et al., 2022). Finally, we take a deep dive into the pre-trained general language models for task-oriented dialogue (TOD), evaluate and compare it with our professionally designed expert model of ZSSF (HiCL). Most researchers will be curious about whether the vanilla capability of pre-trained general TOD models would replace that of all small expert models of ZSSF in this scenario, and whether specially designed generaliza-

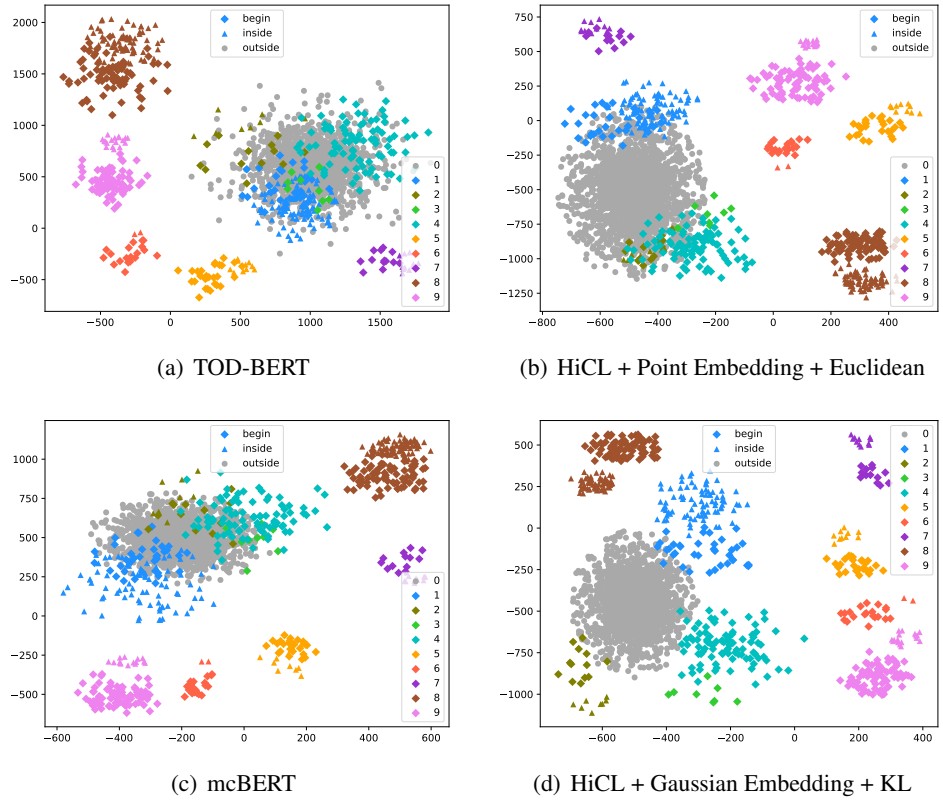

Figure 5: t-SNE visualization in testset of GetWeather target domain on SNIPS dataset for different methods. 1-4 denote unseen slots and 5-9 denote seen slots in all the subfigures.

tion techniques would still work or bring benefits for pre-trained general models in this specific field. This even brings some enlightenment to the research of large language models (LLMs) like Chat-GPT [2], our work bring some explorations into this kind of thoughts. We introduce the pre-training expert model (TOD-BERT) that pre-trained on large corpora of task-oriented dialogue as a baseline to explore that whether this model is good enough for unseen slots generalization, and whether our method can continue to improve the unseen slots performance on top of TOD-BERT, which is also missing from the research (Das et al., 2022).

## A.2 Contrastive Learning in Slot Filling

He et al. (2020) advocate an adversarial attack strengthened contrastive learning with an objective of optimizing the mapping loss from slot entity to slot description in representation space for cross-domain slot filling. Wu et al. (2020) pretrain natural language understanding BERT (Devlin et al., 2019b) for task-oriented dialogue with the joint masked language modeling (MLM) loss

and response contrastive loss (RCL), achieving improvements on slot filling performance for dialogue state tracking. Liu et al. (2021) propose a joint contrastive learning for few-shot intent classification and slot filling in task-oriented dialogue system. Wang et al. (2021) propose a prototypical contrastive learning to bridge semantics gap between token features and slot types in ZSSF. Heo et al. (2022) train BERT encoder with momentum contrastive learning to develop a robust ZSSF model.

In this work, we introduce a Gaussian embedding based hierarchical CL framework. At first, it *coarsely* learns entity-class knowledge of entity type and boundary via entity-level CL. Then, it combines the learned entity-level features, to *finely* learn token-class knowledge of BIO type and boundary via token-level CL. Our method is essentially different from the above approaches.

## B Additional Analysis

### B.1 Performance Gain vs. Dataset Volume

In Tabel 1, 2, 3, 4, we observe that, with the increase of dataset volume (see Table 10), the performance gain of HiCL against baselines gradually

[2]https://openai.com/blog/chatgpt

| Training Setting | | | Zero-shot | | | | | | | | | | | | | | |
|---|---|---|---|---|---|---|---|---|---|---|---|---|---|---|---|---|---|
| | | AP | | BR | | GW | | PM | | RB | | SCW | | FSE | | AVG F1 | |
| Method ⇓ Domain ⇒ | | seen | unseen | seen | unseen | seen | unseen | seen | unseen | seen | unseen | seen | unseen | seen | unseen | seen | unseen |
| *RNN based* | CT† | 47.15 | 5.18 | 51.43 | 1.87 | 39.54 | 2.11 | 46.48 | 0.13 | 25.10 | 0.13 | 39.59 | - | 11.32 | 10.84 | 37.23 | 3.38 |
| | RZT† | 57.50 | 3.48 | 43.84 | 7.14 | 62.84 | 2.34 | 47.45 | 0 | 25.41 | 0.13 | 39.27 | - | 10.61 | 0 | 40.99 | 2.19 |
| | Coach† | 64.65 | 7.94 | 55.88 | 2.89 | 63.97 | 2.20 | 31.69 | 8.78 | 40.42 | 18.81 | 44.35 | - | 27.27 | 15.21 | 46.22 | 9.31 |
| | PCLC† | 73.32 | 2.57 | 62.81 | 16.56 | 65.84 | 14.20 | 45.17 | 17.53 | 34.70 | 25.70 | 53.51 | - | 29.66 | 22.71 | 51.68 | 17.38 |
| | HiCL+BiLSTM (ours) | 65.83 | 17.67 | 54.31 | 14.78 | 66.47 | 14.54 | 46.87 | 16.92 | 40.54 | 19.68 | 54.07 | - | 24.57 | 18.96 | 50.38 | 17.10 |
| *ELMo based* | LEONA‡ | 59.82 | 8.62 | 67.73 | 17.49 | 77.91 | 7.53 | 60.12 | 13.94 | 42.31 | 11.02 | 45.86 | - | 32.16 | 23.43 | 55.13 | 13.67 |
| | HiCL+ELMo (ours) | 65.34 | 17.43 | 71.75 | 15.17 | 73.55 | 18.18 | 58.69 | 18.13 | 40.38 | 20.18 | 49.96 | - | 35.32 | 23.97 | 56.43 | 18.84 |
| *BERT based* | TOD-BERT | 70.27 | 15.97 | 72.57 | 16.75 | 84.12 | 3.03 | 51.63 | 10.79 | 39.57 | 16.42 | 62.85 | - | **46.60** | 38.36 | 61.09 | 16.89 |
| | mcBERT♮ | 77.60 | 17.73 | 81.12 | 34.06 | 91.02 | 10.26 | 52.32 | 10.42 | **68.25** | 10.00 | 70.72 | - | 46.27 | 35.21 | 69.61 | 19.61 |
| | RCSF | | | | | | | | | | | | | | | | 25.44 |
| | HiCL+BERT (ours) | 77.91 | **18.89** | 81.73 | **38.09** | 90.32 | **31.17** | 58.81 | 21.09 | 55.96 | **26.85** | 73.22 | - | 44.65 | **42.15** | 68.94 | **29.71** |
| | w/o coarse CL (ours) | 75.17 | 17.90 | **83.79** | 29.79 | 88.17 | 26.51 | **67.34** | 12.59 | 67.04 | 22.19 | 72.06 | - | 46.31 | 41.63 | **71.41** | 25.10 |
| | w/o fine CL (ours) | **78.29** | 14.18 | 73.41 | 32.92 | **91.95** | 25.82 | 61.41 | **24.31** | 56.86 | 21.15 | 70.63 | - | 42.76 | 35.81 | 67.90 | 25.70 |

Table 6: Detailed F1 scores on SNIPS for seen and unseen slots across all target domains. † denotes the results reported in (Wang et al., 2021). ‡ denotes that we run the publicly released code (Siddique et al., 2021) to obtain the experimental results and ♮ denotes that we re-implemented the model, and we reevaluate their performance on seen and unseen slots following the split method of unseen- and seen-slots sub-test set in Appendix F and the test method of iterative label set semantics inference in Algorithm 2.

| Training Setting | | | Zero-shot | | | | | | | | | | | | |
|---|---|---|---|---|---|---|---|---|---|---|---|---|---|---|---|
| | | AR | | AF | | AL | | FT | | GS | | OS | | AVG F1 | |
| Method ⇓ Domain ⇒ | | seen | unseen | seen | unseen | seen | unseen | seen | unseen | seen | unseen | seen | unseen | seen | unseen |
| *ELMo based* | LEONA‡ | 59.54 | 6.17 | 96.56 | - | 93.11 | - | 85.65 | 3.71 | 51.07 | 31.27 | **84.70** | - | 78.44 | 13.72 |
| | HiCL+ELMo (ours) | 55.34 | 14.81 | 98.34 | - | 93.24 | - | 86.47 | 5.55 | 52.61 | 40.00 | 83.67 | - | 78.28 | 20.12 |
| *BERT based* | TOD-BERT | 47.70 | 9.73 | 96.54 | - | 89.02 | - | 87.09 | 6.45 | 61.30 | 22.22 | 75.80 | - | 76.24 | 12.80 |
| | mcBERT♮ | 66.32 | 9.05 | 98.56 | - | 92.82 | - | 89.26 | 9.20 | 74.58 | 64.44 | 81.35 | - | 83.82 | 27.56 |
| | HiCL+BERT (ours) | **72.51** | **17.65** | 98.09 | - | 92.20 | - | 88.87 | **12.12** | 77.95 | 85.18 | 82.41 | - | **85.34** | **38.32** |
| | w/o coarse CL (ours) | 68.27 | 11.76 | **98.75** | - | **94.20** | - | **90.34** | 9.38 | 69.75 | **85.71** | 71.08 | - | 82.07 | 35.62 |
| | w/o fine CL (ours) | 65.58 | 15.76 | 98.40 | - | 93.51 | - | 89.00 | 10.66 | **81.02** | 83.33 | 84.00 | - | 85.25 | 36.58 |

Table 7: Detailed F1 scores on ATIS for seen and unseen slots across all target domains.

| Training Setting | | | Zero-shot | | | |
|---|---|---|---|---|---|---|
| | | Movie | | Restaurant | | AVG F1 | |
| Method ⇓ Domain ⇒ | | seen | unseen | seen | unseen | seen | unseen |
| *BERT based* | TOD-BERT | 71.72 | - | 56.87 | 0.90 | 64.30 | 0.90 |
| | mcBERT♮ | 76.51 | - | **67.81** | 5.71 | 72.16 | 5.71 |
| | HiCL+BERT (ours) | **77.75** | - | 66.93 | **10.12** | **72.34** | **10.12** |
| | w/o coarse CL (ours) | 74.99 | - | 58.16 | 3.71 | 66.58 | 3.71 |
| | w/o fine CL (ours) | 75.15 | - | 67.51 | 6.34 | 71.33 | 6.34 |

Table 8: Detailed F1 scores on MIT_corpus for seen and unseen slots across all target domains.

| Training Setting | | | Zero-shot | | | | | | | |
|---|---|---|---|---|---|---|---|---|---|---|
| | | BS | | ET | | HE | | RC | | AVG F1 | |
| Method ⇓ Domain ⇒ | | seen | unseen | seen | unseen | seen | unseen | seen | unseen | seen | unseen |
| *BERT based* | TOD-BERT | **45.19** | 27.66 | 75.46 | 21.36 | 81.23 | 58.86 | 56.93 | 50.91 | **64.70** | 39.70 |
| | mcBERT♮ | 25.10 | 28.26 | 68.40 | 13.40 | 81.49 | 47.87 | **59.87** | 55.21 | 58.72 | 36.18 |
| | HiCL+BERT (ours) | 32.83 | 21.23 | 64.84 | 14.28 | 83.00 | 63.69 | 54.66 | 51.78 | 58.83 | 37.75 |
| | HiCL+TOD-BERT (ours) | 35.15 | 22.02 | 75.22 | **23.64** | **83.60** | **90.47** | 55.11 | **59.37** | 62.27 | **48.88** |
| | w/o coarse CL (ours) | 34.54 | 18.64 | 67.68 | 15.87 | 83.43 | 83.70 | 54.35 | 44.87 | 60.00 | 40.77 |
| | w/o fine CL (ours) | 28.69 | **29.97** | 60.59 | 17.51 | 80.53 | 24.76 | 58.85 | 30.42 | 57.17 | 25.67 |

Table 9: Detailed F1 scores on SGD for seen and unseen slots across all target domains.

diminishes. Besides, as indicated in Table 6, 7,8, 9, on the large dataset, HiCL needs to sacrifice more seen slots performance to improve unseen slots performance. This phenomenon indicates that the zero-shot generalization ability of the baseline models gradually becomes stronger with the growth of dataset volume and the diversity of slot types, which helps baseline models to learn more *domain-invariant* slot features for cross-domain ZSSF.

## C  Dataset and Split Details

### C.1  Dataset

We evaluate our approach on four TOD tasks datasets, i.e., SNIPS (Coucke et al., 2018), ATIS (Hemphill et al., 1990), MIT_corpus (Nie et al., 2021) and SGD (Rastogi et al., 2020).

SNIPS (Coucke et al., 2018) is a personal voice assistant dataset that contains 7 domains.

ATIS (Hemphill et al., 1990) is a dataset that contains transcribed audio recordings of people making flight reservations with 18 domains. Domains that contain less than 100 utterances are merged into a single domain Others in our experiments.

MIT_corpus [3] is a spoken query dataset that consists of MIT restaurant domain and MIT movie domain. MIT movie domain contains eng corpus and trivia10k13 corpus, namely simple query version and complex query version, respectively. We merge the two version into one corpus and call it MIT movie domain.

---

[3]The original MIT corpora can be downloaded from https://groups.csail.mit.edu/sls/downloads

SGD (Rastogi et al., 2020) contains 16 domains. However, we find that training on 15 domains except a single target domain, almost all slot types become seen slots. To increase unseen slots and knowledge transfer difficulty, we adopt the same dataset span as (Gupta et al., 2022; Coope et al., 2020) and choose four domains for SGD dataset, namely buses, events, homes, rental cars.

### C.2 Unseen Slots and Seen Slots in Different Domains

Table 12 presents detailed unseen slots and seen slots in different domains for four datasets, i.e., SNIPS, ATIS, MIT_corpus and SGD.

## D Implementation Details

We use uncased BERT[4] to implement the encoder in our model, which has 12 attention heads and 12 transformer blocks. For TOD-BERT, we use the stronger variant[5] that pre-trained using both the MLM and RCL objectives. We uses 100 dimensional Gaussian embeddings, AdamW optimizer (Loshchilov and Hutter, 2019) with $\beta'$ = (0.9, 0.999) and warm-up strategy (warm-up steps is 1% of total training steps). Early stop of patience is set to 30 for stability. $\tau$=0.07 for Eq.(6). dropout rate is 0.3. We set $\alpha$=1 and $\beta$=1 in Eq.(10). We select the best hyperparameters by searching a combination of batch size, learning rate and $\gamma$ in Eq.(10): learning rate is in $\left\{1 \times 10^{-6}, 5 \times 10^{-6}, 1 \times 10^{-5}, 5 \times 10^{-5}\right\}$, batch size is in $\{8, 16, 32, 64\}$, and $\beta$ is in $\{0.001, 0.01, 0.05, 0.1, 0.5\}$. For instance, an optimal learning rate is $1 \times 10^{-5}$ for SNIPS dataset, and $1 \times 10^{-6}$ for MIT_corpus dataset. We select the best-performing model on dev set and evaluate it on test set. We run 5 times for all our experiments and then average them to generate the results. We train and test our model on 4 NVIDIA GeForce RTX 3090 GPUs and 1 NVIDIA Tesla A100 GPU, and it takes averagely less than one hour to reach convergence.

## E Baseline Details

We compare HiCL with the following competing models.

- **Concept Tagger (CT)** (Bapna et al., 2017) is a one-stage leading model for ZSSF, which

adopts slot descriptions to promote the performance on unseen slots in the target domain.

- **Robust Zero-shot Tagger (RZT)** (Shah et al., 2019) incorporates additional slot example entities combined with slot descriptions to improve zero-shot adaption.

- **Coarse-to-fine Approach (Coach)** (Liu et al., 2020) is a pioneer of two-stage framework for ZSSF, which divides the ZSSF task into two stages: coarse-grained slot entity segmentation in the form of BIO and fine-grained alignment between slot entities and slot types by utilizing slot descriptions. We use their stronger variant Coach+TR and call it Coach for brevity.

- **Contrastive Zero-Shot Learning with Adversarial Attack (CZSL-Adv)** (He et al., 2020) is an improver of Coach, which employs contrastive learning and adversarial attack training to optimize the performance of the framework.

- **Prototypical Contrastive Learning and Label Confusion (PCLC)** (Wang et al., 2021) is a two-stage based approach, which employs prototypical contrastive learning and label confusion strategy to enhance the robustness of unseen slots filling under zero-shot setting.

- **Linguistically-Enriched and Context-Aware (LEONA)** (Siddique et al., 2021) is an advocate of three-stage ZSSF model, which utilizes context-aware and linguistic token representation to improve the effect on semantic similarity modeling between utterance tokens and slot descriptions based on attention mechanism.

- **Reading Comprehension for Slot Filling (RCSF)** (Yu et al., 2021) is the current question answering (QA) based SOTA model, which formulate ZSSF task as a machine reading comprehension (MRC) problem.

- **Momentum Contrastive Learning with BERT(mcBERT)** (Heo et al., 2022) is current state-of-the-art model (to our knowledge), which improves ZSSF performance by adopting BERT backbone and training it with momentum contrastive learning.

---

[4] https://huggingface.co/bert-base-uncased
[5] https://huggingface.co/TODBERT/TOD-BERT-JNT-V1

- **TOD-BERT** (Wu et al., 2020) Pre-trained Natural Language Understanding for Task-Oriented Dialogue. This model is current state-of-the-art pre-training model for TOD, we use their stronger variant that pre-trained with the joint technique of masked language modeling (MLM) loss and response contrastive loss (RCL), and name it TOD-BERT for brevity.

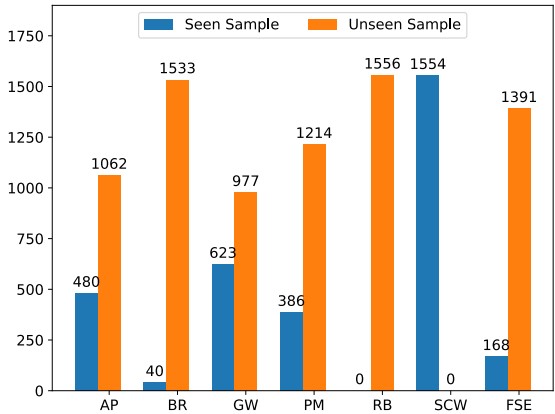

Figure 6: Unseen and seen slots test set split on SNIPS dataset with the method of Coach (Liu et al., 2020). The figure on the top of bar chart denotes the number of seen or unseen samples for different target domain in test set. "0" represents no sample (unseen or seen) exists.

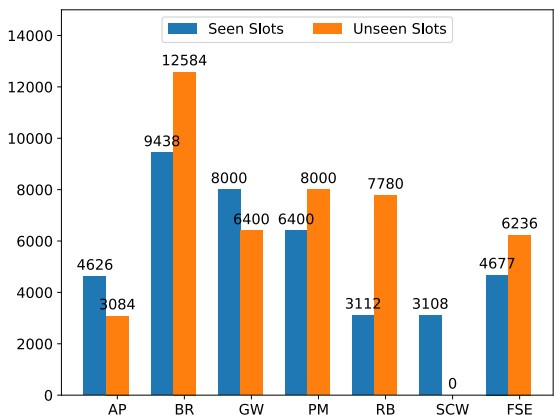

Figure 7: Unseen and seen slots test set split on SNIPS dataset with our method. The figure on the top of bar chart denotes the number of seen or unseen slots for different target domain in test set. "0" represents no unseen-slot exists.

## F    Rectified Test Set Split for Unseen Slots and Seen Slots

In the cross-domain slot filling scenario, seen slots refer to those slot types that appear in both source domains and target domain, while unseen slots refer to those slot types that only appear in target domain. As shown in Figure 6, Liu et al. (2020) divide SNIPS (Coucke et al., 2018) test set into **unseen** and **seen** subset according to whether an utterance contains at least one unseen slot, which leads to unseen-seen slots overlapping problems in unseen slots performance evaluation. Since the performance results of the unseen samples are actually entangled with seen slots (unseen sample test set contains both unseen slots and seen slots), which seriously causes a bias in testing a model's actual performance on unseen slots.

For example, for an utterance "will it be colder in connorville" in target test set, its corresponding slot label is "O O O B-condition_temperature O B-city", this utterance sample comprises both unseen slot type *condition_temperature* and seen slot type *city*. Nevertheless, this sample will be classified into unseen test set according to the method of Coach (Liu et al., 2020).

We rectify this bias by splitting unseen and seen test set with slot granularity strategy instead of sample granularity method used in Coach (Liu et al., 2020). This slot granularity split method is illustrated in § 4.5, Algorithm 1 and 2 (iterative label set semantics inference), which is able to train and test unseen and seen slots, separately and unbiasedly. For instance, following this new split method, as shown in Figure 7, slot type *condition_temperature* and *city*, will be re-classified into unseen subset and seen subset for the utterance "will it be colder in connorville" , respectively (§ 4.5). Through the comparison between Figure 6 and Figure 7, it is observed that unseen and seen sub-test set across domains distribute more evenly by employing our approach over Coach (Liu et al., 2020) method. Furthermore, we can find the defect of original Coach (Liu et al., 2020) split method. For example, in target domain RateBook, as shown in 6, the split result indicates that the number of seen samples (or seen slots) is zero and all samples are unseen ones. However, splitting with our new method, the number of seen slots is 3,112 and the number of unseen slots is 7,780. We can see approximately one third of slots in domain RateBook are seen slots, but they were wrongly classified into unseen test set by this biased split method of Coach (Liu et al., 2020), and their performance was regarded as that of unseen samples. To our knowledge, almost all baselines follow the split method of Coach (Liu

et al., 2020), so we have to reevaluate their real performance for unseen slots with our new split method, i.e., iterative label set semantics inference illustrated in § 4.5, Algorithm 1 and 2.

| Dataset | Query Numbers | | | | |
|---|---|---|---|---|---|
| | train | val | test | seen slots | unseen slots |
| SNIPS | 110,007 | 27,519 | 82,488 | 38,917 | 46,651 |
| ATIS | 342,574 | 85,741 | 256,833 | 216,147 | 40,817 |
| MIT_corpus | 1,464,532 | 365,433 | 1,099,099 | 935,265 | 163,853 |
| SGD | 2,780,748 | 695,285 | 2,085,463 | 1,367,438 | 718,025 |

Table 10: Data statistics of training, validation and test for all domains of SNIPS, ATIS, MIT_corpus and SGD, respectively, after data augmentation. All baselines and HiCL are fine-tuned on the same augmented dataset.

| Gold | HiCL | mcBERT |
|---|---|---|

| | *test sample* ⇒ | book indian food at a highly rated pub for 1 for 02:22 pm | |

***unseen slots vs. slot entities***

| Gold | HiCL | mcBERT |
|---|---|---|
| restaurant_name : None | restaurant_name : None | restaurant_name : None |
| facility : None | facility : None | facility : None |
| cuisine : indian | cuisine : indian | cuisine : None |
| restaurant_type : pub | restaurant_type : pub | restaurant_type : pub |
| served_dish : None | served_dish : None | served_dish : None |
| party_size_number : 1 | party_size_number : None | party_size_number : None |
| poi : None | poi : None | poi : None |
| party_size_description : None | party_size_description : None | party_size_description : None |

***seen slots vs. slot entities***

| Gold | HiCL | mcBERT |
|---|---|---|
| city : None | city : None | city : None |
| timerange : 02:22 pm | timerange : 1 02:22 pm | timerange : 02:22 pm |
| country : None | country : None | country : indian |
| sort : highly rated | sort : highly rated | sort : highly rated |
| spatial_relation : None | spatial_relation : None | spatial_relation : None |
| state : None | state : None | state : None |

| | *test sample* ⇒ | i d like a table for ten in 2 minutes at french horn sonning eye | |

***unseen slots vs. slot entities***

| Gold | HiCL | mcBERT |
|---|---|---|
| restaurant_name : french horn sonning eye | restaurant_name : french horn sonning eye | restaurant_name : french horn sonning eye |
| facility : None | facility : None | facility : french horn sonning eye |
| cuisine : None | cuisine : None | cuisine : french sonning |
| restaurant_type : None | restaurant_type : None | restaurant_type : None |
| served_dish : None | served_dish : None | served_dish : None |
| party_size_number : ten | party_size_number : None | party_size_number : None |
| poi : None | poi : None | poi : french horn sonning eye |
| party_size_description : None | party_size_description : None | party_size_description : None |

***seen slots vs. slot entities***

| Gold | HiCL | mcBERT |
|---|---|---|
| city : None | city : None | city : None |
| timerange : in 2 minutes | timerange : in 2 minutes | timerange : ten in 2 minutes |
| country : None | country : None | country : None |
| sort : None | sort : None | sort : None |
| spatial_relation : None | spatial_relation : None | spatial_relation : None |
| state : None | state : None | state : None |

| | *test sample* ⇒ | how cold will it be here in 1 second | |

***unseen slots vs. slot entities***

| Gold | HiCL | mcBERT |
|---|---|---|
| condition_temperature : cold | condition_temperature : cold | condition_temperature : cold |
| current_location :here | current_location : None | current_location : None |
| condition_description : None | condition_description : None | condition_description : None |
| geographic_poi : None | geographic_poi : None | geographic_poi : None |

***seen slots vs. slot entities***

| Gold | HiCL | mcBERT |
|---|---|---|
| timerange : in 1 second | timerange : in 1 second | timerange : in 1 second |
| state : None | state : None | state : None |
| city : None | city : None | city : None |
| country : None | country : None | country : None |
| spatial_relation : None | spatial_relation : None | spatial_relation : None |

| | *test sample* ⇒ | will it get warmer in czechia | |

***unseen slots vs. slot entities***

| Gold | HiCL | mcBERT |
|---|---|---|
| condition_temperature : warmer | condition_temperature : warmer | condition_temperature : warmer |
| current_location : None | current_location : None | current_location : None |
| condition_description : None | condition_description : None | condition_description : None |
| geographic_poi : None | geographic_poi : None | geographic_poi : None |

***seen slots vs. slot entities***

| Gold | HiCL | mcBERT |
|---|---|---|
| timerange : None | timerange : None | timerange : None |
| state :None | state :None | state :None |
| city : None | city : None | city : None |
| country : czechia | country : czechia | country : czechia |
| spatial_relation : None | spatial_relation : None | spatial_relation : None |

| | *test sample* ⇒ | tell me if it ll be freezing next month in rhode island | |

***unseen slots vs. slot entities***

| Gold | HiCL | mcBERT |
|---|---|---|
| condition_temperature : freezing | condition_temperature : freezing | condition_temperature : None |
| current_location : None | current_location : None | current_location : None |
| condition_description : None | condition_description : None | condition_description : None |
| geographic_poi : None | geographic_poi : None | geographic_poi : None |

***seen slots vs. slot entities***

| Gold | HiCL | mcBERT |
|---|---|---|
| timerange : next month | timerange : next month | timerange : next month |
| state : rhode island | state : rhode island | state : rhode island |
| city : None | city : None | city : None |
| country : None | country : None | country : None |
| spatial_relation : None | spatial_relation : None | spatial_relation : None |

| | *test sample* ⇒ | when will it be temperate in lansford | |

***unseen slots vs. slot entities***

| Gold | HiCL | mcBERT |
|---|---|---|
| condition_temperature : temperate | condition_temperature : temperate | condition_temperature : None |
| current_location : None | current_location : None | current_location : None |
| condition_description : None | condition_description : None | condition_description : None |
| geographic_poi : None | geographic_poi : None | geographic_poi : None |

***seen slots vs. slot entities***

| Gold | HiCL | mcBERT |
|---|---|---|
| timerange : None | timerange : None | timerange : None |
| state : None | state : None | state : None |
| city : lansford | city : lansford | city : lansford |
| country : None | country : None | country : None |
| spatial_relation : None | spatial_relation : None | spatial_relation : None |

| | *test sample* ⇒ | is maalaea has chillier weather | |

***unseen slots vs. slot entities***

| Gold | HiCL | mcBERT |
|---|---|---|
| condition_temperature : chillier | condition_temperature : chillier | condition_temperature : None |
| current_location : None | current_location : None | current_location : None |
| condition_description : None | condition_description : None | condition_description : None |
| geographic_poi : None | geographic_poi : None | geographic_poi : None |

***seen slots vs. slot entities***

| Gold | HiCL | mcBERT |
|---|---|---|
| timerange : None | timerange : None | timerange : None |
| state : None | state : None | state : None |
| city : maalaea | city : maalaea | city : None |
| country : None | country : None | country : None |
| spatial_relation : None | spatial_relation : None | spatial_relation : None |

Table 11: Iterative label set semantics inference (ILSSI) prediction examples from SNIPS dataset with BookRestaurant, GetWeather as target domain, respectively.

| | SNIPS | |
|---|---|---|
| **Target Domain** | **Unseen Slots** | **Seen Slots** |
| AddToPlaylist | entity_name, playlist_owner | artist, playlist, music_item |
| BookRestaurant | party_size_description, restaurant_type, poi, served_dish, party_size_number, cuisine, facility, restaurant_name | city, spatial_relation, state, sort, timeRange, country |
| GetWeather | condition_description, current_location, condition_temperature, geographic_poi | country, spatial_relation, state, timeRange, city |
| PlayMusic | track, service, album, year, genre | artist, music_item, sort, playlist |
| RateBook | object_select, rating_unit, rating_value, best_rating, object_part_of_series_type | object_name, object_type |
| SearchCreativeWork | - | object_type, object_name |
| FindScreeningEvent | object_location_type, movie_type, location_name, movie_name | object_type, timeRange, 'spatial_relation' |
| | **ATIS** | |
| **Target Domain** | **Unseen Slots** | **Seen Slots** |
| Abbreviation | booking_class, meal_code, days_code | meal, airline_code, class_type, airport_code, mod, fromloc.city_name, aircraft_code, fare_basis_code, toloc.city_name', restriction_code, airline_name |
| Airfare | - | flight_number, fare_amount, arrive_date.date_relative, flight_time, cost_relative, fromloc.city_name, return_date.day_number, toloc.city_name, depart_date.day_number, toloc.state_name, flight_mod, arrive_date.day_name, depart_date.date_relative, connect, depart_time.time_relative, stoploc.city_name, toloc.airport_name, airline_name, arrive_time.time_relative, arrive_date.day_number, economy, arrive_time.time, depart_time.period_mod, depart_date.day_name, return_date.date_relative, return_date.month_name, aircraft_code, meal, fromloc.airport_code, arrive_date.month_name, fromloc.state_name, flight_stop, or, fromloc.airport_name, depart_date.month_name, toloc.state_code, depart_time.time, class_type, airline_code, toloc.airport_code, round_trip, depart_date.today_relative, depart_time.period_of_day, return_date.day_name, fromloc.state_code, depart_date.year, flight_days |
| Airline | - | depart_time.time, mod, flight_stop, toloc.state_code, fromloc.city_name, city_name, airport_name, flight_days, flight_number, arrive_time.period_of_day, connect, airline_code, cost_relative, depart_time.period_of_day, fromloc.airport_code, fromloc.airport_name, depart_time.start_time, arrive_date.month_name, depart_time.end_time, arrive_time.time, toloc.city_name, class_type, fromloc.state_code, toloc.airport_name, depart_date.day_number, depart_date.month_name, airline_name, depart_time.time_relative, arrive_date.day_number, depart_date.day_name, stoploc.city_name, round_trip, depart_date.today_relative, depart_date.date_relative, toloc.state_name, aircraft_code |
| Flight | stoploc.airport_code, stoploc.state_code, flight, return_time.period_mod, compartment, return_time.period_of_day, toloc.country_name, arrive_time.end_time, stoploc.airport_name, arrive_time.period_mod, arrive_date.today_relative, return_date.today_relative, arrive_time.start_time | mod, fare_amount, depart_date.month_name, day_name, airport_name, toloc.city_name, arrive_time.time_relative, cost_relative, flight_time, flight_number, flight_mod, period_of_day, or, fare_basis_code, depart_date.year, fromloc.city_name, depart_date.day_name, toloc.airport_code, return_date.date_relative, arrive_date.date_relative, fromloc.airport_name, class_type, meal_description, depart_date.date_relative, depart_time.period_mod, toloc.state_code, flight_days, return_date.day_number, depart_date.day_number, economy, arrive_time.period_of_day, flight_stop, meal, aircraft_code, depart_time.time, toloc.state_name, depart_date.today_relative, depart_time.end_time, airport_code, airline_name, city_name, return_date.month_name, round_trip, arrive_time.time, arrive_date.day_number, return_date.day_name, depart_time.time_relative, arrive_date.month_name, airline_code, connect, depart_time.start_time, toloc.airport_name, depart_time.period_of_day, fromloc.state_code, fromloc.state_name, arrive_date.day_name, fromloc.airport_code, stoploc.city_name |
| Ground Service | day_number, today_relative, time, time_relative, month_name | depart_date.day_name, or, fromloc.city_name, state_name, depart_date.day_number, day_name, toloc.airport_name, airport_code, depart_date.date_relative, flight_time, state_code, city_name, depart_date.month_name, toloc.city_name, airport_name, period_of_day, transport_type, fromloc.airport_name |
| Others | - | airport_name, flight_time, toloc.state_code, fromloc.state_name, depart_date.day_number, round_trip, flight_number, airport_code, fromloc.airport_name, fare_amount, flight_days, toloc.airport_name, stoploc.city_name, toloc.city_name, depart_date.today_relative, transport_type, economy, aircraft_code, toloc.state_name, arrive_date.month_name, cost_relative, city_name, restriction_code, toloc.airport_code, flight_mod, state_code, fromloc.airport_code, mod, meal, depart_date.date_relative, meal_description, depart_date.month_name, arrive_time.time_relative, arrive_date.day_number, airline_code, depart_time.time, depart_date.day_name, depart_time.time_relative, arrive_date.day_name, class_type, or, fromloc.city_name, arrive_time.time, flight_stop, fare_basis_code, state_name, airline_name, depart_time.period_of_day |
| | **MIT_corpus** | |
| **Target Domain** | **Unseen Slots** | **Seen Slots** |
| Movie | - | director, ratings_average, plot, rating, title, trailer, actor, review, year, genre, character, song, quote, opinion, award, origin, soundtrack, character_name, relationship |
| Restaurant | location, dish, hours, cuisine, restaurant_name, price, amenity | year, director, actor, song, title, rating, quote, trailer, plot, opinion, review, origin, genre, character, soundtrack, relationship, character_name, ratings_average, award |

| | SGD | |
|---|---|---|
| **Target Domain** | **Unseen Slots** | **Seen Slots** |
| Buses | origin_airport, destination_city, origin_city, from_station, to_location, destination_airport, fare, origin, from_city, from_location, origin_airport_name, origin_station_name, to_city, outbound_arrival_time, departure_time, destination_airport_name, outbound_departure_time, to_station, destination_station_name, leaving_time, inbound_departure_time | venue, percent_rating, street_address, price, address, destination, movie_title, leaving_date, alarm_time, date, return_date, pickup_city, time, average_rating, rating, alarm_name, attraction_name, phone_number, check_in_date, city, event_name, start_date, hotel_name, pickup_date, place_name, category, price_per_ticket, venue_address, departure_date, price_per_night |
| Events | genre, cast, humidity, temperature, event_date, starring, city_of_event, director, available_start_time, address_of_location, title, event_location, available_end_time, wind, subcategory, cuisine, artist, account_balance, album, precipitation, song_name, directed_by, aggregate_rating | rating, movie_title, destination, balance, wait_time, leaving_date, attraction_name, hotel_name, return_date, venue_address, pickup_date, ride_fare, end_date, date, departure_date, time, city, phone_number, category, pickup_time, price_per_day, street_address, venue, start_date, event_name, approximate_ride_duration, movie_name, address, average_rating, price_per_ticket, pickup_location, percent_rating, price_per_night, place_name |
| Homes | area | dropoff_date, start_date, total_price, ride_fare, destination, pickup_time, pickup_location, phone_number, address, balance, rent, price, pickup_date, alarm_time, approximate_ride_duration, alarm_name, visit_date, wait_time, property_name |
| Rental Cars | check_out_date, location, car_name | pickup_city, total_price, pickup_date, start_date, city, price_per_night, dropoff_date, average_rating, end_date, rent, destination, pickup_location, phone_number, attraction_name, address, property_name, movie_name, pickup_time, street_address, hotel_name, check_in_date, price_per_day, leaving_date, visit_date, rating |

Table 12: Unseen and seen slots in different domains of SNIPS, ATIS, MIT_corpus, and SGD dataset, respectively. The evaluation paradigm is that adopting each single domain as target or test domain and the remainder of domains as training domains in the dataset. '-' denotes no slot type exists.