# OpenReview forum: "HierarchicalContrast: A Coarse-to-Fine Contrastive Learning Framework for Cross-Domain Zero-Shot Slot Filling"
_EMNLP/2023/Conference — EMNLP 2023 Findings_

### Official Review · Reviewer_8SoH · 2023-08-02

**Typos Grammar Style And Presentation Improvements:** 1. The table design and style of the …
**Soundness:** 3

**Excitement:**

3: Ambivalent: It has merits (e.g., it reports state-of-the-art results, the idea is nice), but there are key weaknesses (e.g., it describes incremental work), and it can significantly benefit from another round of revision. However, I won't object to accepting it if my co-reviewers champion it.

**Missing References:**

[1] Cross-Domain Slot Filling as Machine Reading Comprehension.

**Paper Topic And Main Contributions:**

This paper proposes a hierarchical contrastive learning based method to alleviate the domain shift problem under the cross domain setting.  Extensive experiments are conducted to verify the effectiveness of the proposed method.

**Questions For The Authors:**

1. Is the loss jointly optimized or cross staged as Figure 3 described?
2. How do the authors get entity level representations in the Coarse training stage? Is it averaged over tokens within a single entity?

**Reasons To Accept:**

The experiments in this paper are solid under several settings and good performance is achieved over previous baseline methods.
Moreover, the authors have made fair comparison based on different pretrained backbones.

**Reasons To Reject:**

1. Lack of MRC-based SF baselines under the same setting, such as  [1]
2. The description of method is unclear and it is not easy to follow, such as, how to get y_i in loss computation (Eq.9) is incomplete, which means the CRF seems untrained.
3. The design of contrastive loss based on Gaussian embeddings seems to be borrowed from Das et al., 2022. But the authors do not cite, explain or clarify it in the methodology section.

[1] Cross-Domain Slot Filling as Machine Reading Comprehension.

**Reproducibility:**

4: Could mostly reproduce the results, but there may be some variation because of sample variance or minor variations in their interpretation of the protocol or method.

**Reviewer Confidence:**

4: Quite sure. I tried to check the important points carefully. It's unlikely, though conceivable, that I missed something that should affect my ratings.

---

> ### Author Rebuttal · Authors · 2023-08-23
>
> Thanks for your review.
> 1. For question 1, in implementation, we first update the gradient according to coarse-grained contrastive loss (entity-level),  and then update the gradient according to the combination loss of fine-grained contrastive loss (token-level) and loss of CRF,  however, we empirically found that there is little performance difference between simultaneously gradients update and sequentially gradients update for coarse-grained contrastive learning loss , fine-grained contrastive learning loss  and CRF loss via additional experiments that did not present in our paper.
>
> &emsp;&emsp;The implemented codes are listed below：
>
>             crf_loss, coarse_loss, fine_loss, _ = model(features)
>             coarse_loss = coarse_loss.mean()
>             coarse_losses.append(coarse_loss.detach().cpu().item())
>             loss = crf_loss + model_args.beta*fine_loss
>             loss = loss.mean()
>             losses.append(loss.detach().cpu().item())
>
>             # first step
>             optim.zero_grad()
>             coarse_loss.backward(retain_graph=True)
>             optim.step()
>             scheduler.step()
>
>             # second step
>             optim.zero_grad()
>             loss.backward(retain_graph=True)
>             optim.step()
>             scheduler.step()
>             pbar.set_description(f"COARSE_LOSS: {coarse_losses[-1]:.4f}  LOSS: {losses[-1]:.4f}")
> 2.  For question 2, actually both coarse-grained (entity-level) contrastive training and fine-grained (token-level) contrastive training share the same representations of utterances, the difference is their slot labels, e.g, entity-level slot label set is {'O': 0, 'date_time': 1, 'location': 2 } and token-level slot label set is {'O': 0, 'B-date_time': 1, 'I-date_time': 2, 'B-location': 3, 'I-location': 4 }.
> This representations of utterances  are obtained by firstly encoding utterances token by token with BERT,  and then transforming the  point embeddings of BERT output into Gaussian distributed embeddings.
>
> As to reasons to reject, we give the following supplements and explanations:
> 1.  We constrain our work and research scope in task-oriented dialogue field, therefore the baselines and datasets for comparison and evaluation are selected from   task-oriented dialogues ( including spoken language understanding sub-field ), the research scope was stated in  abstract(line 001),  introduction(line 030), contribution(line 201), Future work(line 583).
> 2. $y_i$ is predicted slot labels and obtained from BERT model output (point embedding) by encoding utterances and then compute the CRF loss (negative log likelihood) with ground truth through Viterbi decoding.
> For slot label prediction and compute CRF loss in Eq.9,  they are basically the same as previous SOTA models, like Coach , LEONA or mcBERT, which are not our major innovation and less described in context.
> 3. For the similar technique application of Gaussian embedding  in Das et al., 2022, we cited this paper in introduction part (line 122 and line 123) and related work (appendix A.1, line 867-line 869).  If you mean Eq.(2), we could clarify this Gaussian embedding transformation formulation appears even more earlier in several previous literatures(e.g., computer vision paper), it is not initially originated from Das et al., 2022.
> And if we implement the Gaussian transformation by referring to the codes of this paper (Das et al., 2022 ), we will clarify it when we release our code in GitHub.
>
> 4. Besides, the major difference our method and the method of Das et al., 2022 are below:
>
> &emsp;&emsp;a. We design a hierarchical contrastive learning (entity-level and token-level, and empirically verify that they complement each other and better than employing only one of them) , while the method of Das et al., 2022 only present the idea of entity-level contrastive learning（the tokens in this paper actually denotes entities). So our paper represents a first of its kind of token-level contrastive learning and
> is the originator and pioneer of proposing the method of token-level contrastive learning.
>
> &emsp;&emsp;b. The second difference is that we focus on improve the unseen slots performance rather than both (unseen slots and seen slots), which did not refer or involve in the work of Das et al., 2022, it did not classify the entities into unseen ones and seen ones.
>
>  &emsp;&emsp;c. The research area is very different, we study the field of task-oriented dialogues  (single-turn and multi-turn dialogues datasets),  which inherently has gigantic distinction with other kind of NER tasks, like OntoNotes or CoNLL-2003.
>
> &emsp;&emsp;d. We study in-depth the pre-trained general language model in task-oriented dialogue field and evaluate and compare it with our professionally designed expert model of zero-shot slot filling, which brings some enlightenment even for large language models （LLMs）like ChatGPT, most researchers will be curious about whether vanilla capability of LLMs would replace all small expert models in this scenarios, and whether specially designed generalization techniques still work or bring benefits for pre-trained general models in specific field, our work bring some explorations for this kind of thoughts.
> We introduce the pre-training expert model (TOD-BERT) as baseline to explore that whether the pre-training model that pre-trained on large corpora of task-oriented dialogue (TOD) are good enough for unseen slots generalization, and whether our method can continue to improve the unseen slots performance on top of it,  this research is also missing in Das et al., 2022.

---

### Official Review · Reviewer_ohcU · 2023-08-05

**Typos Grammar Style And Presentation Improvements:** The author should reorganize the stru…
**Soundness:** 2

**Excitement:**

3: Ambivalent: It has merits (e.g., it reports state-of-the-art results, the idea is nice), but there are key weaknesses (e.g., it describes incremental work), and it can significantly benefit from another round of revision. However, I won't object to accepting it if my co-reviewers champion it.

**Paper Topic And Main Contributions:**

This paper proposes a hierarchical CL approach to discover new slot entities for Cross-Domain Zero-Shot Slot Filling. They also use Gaussian embedding to represent token features.

**Reasons To Accept:**

* The idea is simple and reasonable.

**Reasons To Reject:**

* The presentation of this paper is a little rough. The Introduction part is too long, and many important results and analysis are in the appendix. The author should reorganize the structure of the paper.
* The results on SNIPS and SGD seem comparable to previous methods.

**Reproducibility:**

3: Could reproduce the results with some difficulty. The settings of parameters are underspecified or subjectively determined; the training/evaluation data are not widely available.

**Reviewer Confidence:**

4: Quite sure. I tried to check the important points carefully. It's unlikely, though conceivable, that I missed something that should affect my ratings.

---

> ### Author Rebuttal · Authors · 2023-08-23
>
> Thanks for the review.  Just a kind reminder, please review more impartially, professionally and conscientiously.
>
> 1. Long introduction part aims to help non-expert reviewers to understand the concepts which are difficult to follow, like
> slot filling, slot type, slot entity, etc, more importantly, the background and  the downside or weakness occurred in zero-shot slot filling task which we aims to tackle, that is important because it give the rationales why we propose hierarchical contrastive learning and Gaussian embedding to improve the generalization ability ( unseen slots performance ) of the zero-shot slot filling models. In other words, why we do not use other methods or techniques to improve unseen slots performance (unseen slots generalization ability)?
>
> 2. Please point specific claims/arguments that are not sufficiently supported in our paper,  we read carefully for our paper several times again. and find that our claims in the paper is sufficiently and comprehensively supported both qualitatively and quantitatively, for example,  t-SNE visualization and  the detailed case studies, and the extremely detailed and thoroughly diverse ablation studies.
>
> 3. Our HiCL mainly focus on big improvement of unseen slots performance and maintain the overall performance of zero-shot slot filling.
> Compared with all baselines, our HiCL improve averaged unseen slots performance by at least 10.1% (absolute value, see table 5) and 9.18% (absolute value, see table 8) for SNIPS and SGD dataset, respectively, which is very significantly and unprecedentedly. Besides the overall performance ( the averaged performance for both unseen and seen slots ) is comparable or even better than existing SOTA models.
>
> 4. The previously reported results of the SOTA models for both unseen slots performance and  total performance is biased and exaggerated.  The reasons are:
>
>  &emsp;&emsp;a. the existing methods regards a significant parts of seen slots performance as unseen slots performance, which overstated the actual effect of unseen slots evaluation/reported results in their papers.
>
>  &emsp;&emsp;b. In inference phase, the previous SOTA models only test one or two negative samples (if a given test utterance does not include certain slot type, this sample is negative sample, each sample contains slot_type+utterance) ,  which will results in a higher reported results than actual results,  while we test all negative samples (test all slot types pre-defined in slot type set that are NOT actually appear in test utterance) and results in a more objective overall performance results.
>
> &emsp;&emsp;c. In practice,  in our iterative label inference method ( described in table 11),  all slot types should indeed be tested one by one (including all negative samples ) for a given utterance in model  inference,   because you never know which slot type exists in one given utterance and which does not exist in advance, which is different from traditional models (like Coach) that predict all slot types once time when given an utterance.
>
> &emsp;&emsp; In summary, we unbiasedly and correctly re-evaluate the real performance (unseen slots and overall slots performance) both baselines and our methods.

---

### Official Review · Reviewer_GdPM · 2023-08-10

**Typos Grammar Style And Presentation Improvements:** You should resize the tables and pict…
**Soundness:** 1

**Excitement:**

2: Mediocre: This paper makes marginal contributions (vs non-contemporaneous work), so I would rather not see it in the conference.

**Missing References:**

Du X, He L, Li Q, et al. QA-driven zero-shot slot filling with weak supervision pretraining[C]//Proceedings of the 59th Annual Meeting of the Association for Computational Linguistics and the 11th International Joint Conference on Natural Language Processing (Volume 2: Short Papers). 2021: 654-664.

**Paper Topic And Main Contributions:**

The paper proposes a coarse- to fine-grained contrastive learning based on Gaussian-distributed embedding to learn the generalized semantic relationship between utterance-tokens. Experiments on two datasets show that the proposed framework achieve SOTA performance.

**Questions For The Authors:**

1. What is  $n_u$ in Eq.6?
2. Could you introduce more details in Figure 4 and Figure 5?

**Reasons To Accept:**

1. The proposed framework achieves SOTA performance.
2. The paper find unseen slots and seen slots overlapping problem in test set split, rectify the bias by splitting from slot granularity.
3. The paper conduct experiments on four datasets.

**Reasons To Reject:**

1. In my opinion, the paper needs to be rewritten.The article has too much content that is placed in the appendix, which is unreasonable. Moreover, many tables and pictures（Figure 2,3 Table 1,2,3） in the article are too small, making it difficult for readers to see the internal details clearly.
2. The method is not novel. Contrastive learning has been widely used in slot filling tasks, although the task is not exactly the same. I think the novelty of this simple modification is not suitable for EMNLP.
3. Many variables (em. $n_u$ in Eq.6) in the formula lack corresponding explanations. I suggest you re-examine the paper carefully and add these explanations.

**Reproducibility:**

2: Would be hard pressed to reproduce the results. The contribution depends on data that are simply not available outside the author's institution or consortium; not enough details are provided.

**Reviewer Confidence:**

4: Quite sure. I tried to check the important points carefully. It's unlikely, though conceivable, that I missed something that should affect my ratings.

---

> ### Author Rebuttal · Authors · 2023-08-23
>
> 1. for $n_u$ in Eq.6, we have already presented the explanation and definition of $n_u$ in line 303 and line 304 of our paper.
>
> 2. In Figure 4 and 5,  we provide a visualization to illustrate the effectiveness of our proposed hierarchical contrastive learning(CL), i.e.,  both entity-level CL and token-CL play a critical contribution for the final big performance improvement of unseen slots across various target domains,  while generally they do not hurt the seen slots performance in task-oriented dialogue.
>
> 3. The pictures are saved in SVG format and you can enlarge the PDF file in any size without any loss on fidelity.
>
> $\textbf{We extremely hard to believe and understand the justice of the scores !}$
> $\textbf{please more respect the works of authors,}$
> $\textbf{please be more professional and qualified for the reviews, }$
> $\textbf{and please be more known about the innovation and contribution of our paper.}$
>
> $\textbf{Have you read carefully our paper?  Please list your solid and convincing reasons for your scores.}$
>
> For novelty, we introduce hierarchical contrastive learning (entity level and token-level) which is the first time study in slot filling of task-oriented dialogue field, moreover, we employ Gaussian embedding to boost the robustness of slot filling models, which is also the first time exploration.
>
> For the problem to solve, we aims to improve the unseen slot performance for current zero-shot slot filling models in task-oriented dialogue, whose real performance is not correctly evaluated previously , which is not thoroughly studied and has been proved to have a lot of room for improvement.
>
> For experiments, we spend more than half year to do various  experiments and ablation analysis.
>
> You should understand whether the problem raised is convincing, why we use entity-level CL, token-level CL and Gaussian Embedding, how we rectify the evaluation for the unseen slot performance, and so on.

---

### Official Review · Reviewer_vCNz · 2023-08-11

**Soundness:** 3

**Excitement:**

4: Strong: This paper deepens the understanding of some phenomenon or lowers the barriers to an existing research direction.

**Paper Topic And Main Contributions:**

This paper discovers the problem that existing methods on zero-shot slot filling task have poor performance on unseen target domains due to their poor generalizability. They propose a novel hierarchical contrastive learning (CL) approach based on Gaussian embedding to learn and extract slot-agnostic features across utterance-tokens. The main contributions are firstly, they add token-level CL besides entity-level CL and secondly, they use Gaussian embedding so that each token becomes a density instead of a single point. Both innovations help to learn the latent relations between tokens (e.g, if two tokens share the same slot type), in order to improve the generalizability in unseen domains. The results show that the improvement on unseen dataset is non-trivial and the paper conduct comprehensive ablation studies to discuss each component's impact.



**Questions For The Authors:**

Question A:  When you calculate coarse-grained loss, how do the slot types encoded? For example, ‘this week’ is date_time, date_time, or ‘this week’ is trained as a whole?

Question B: in the figure 3, it shows that a sequence of slot types and two utterances are sent into encoders? Are those slot types related to every token/entity in the two utterances? In the Gaussian Transformation Network, the slot types are also trained? Also in the output, the slot labels are entity-level or token-level? Please illustrate more on the figure.

Question C: why only test on one unseen target domain? Did you try other target domains ? Usually, the model should be tested on several target domains to prove generalizability?

Question D: what is intra-token? tokens in the same entity?


**Reasons To Accept:**

1. The paper is quite comprehensive in introduction, problem definition, experiments and ablation studies. The author illustrates the problem and different paradigms of zero-shot slot fillings very clearly. It is good for people who are not experts in this field to learn the concept quickly.

2. The idea of adding token-level CL is innovative. It is not very intuitive but the author explains it quite well and the results show the great performance of the framework.

3. The ablation studies are detailed. The author discusses about the components from different aspects. The ablation studies make this work more convincing.

**Reasons To Reject:**

The method part needs more clarification (more details in the questions part).

In the contribution part, the author said the model is to extract self-agnostic features across tokens. However, in the analysis part, they said they hypothesize the model have learned self-agnostic features. It is kind of self-contradictory since the author cannot know if the model really learn self-agnostic features. The concept of self-agnostic sounds intriguing but also abstract. It would be better if author can conduct more analysis on if the relations between tokens are learned.

**Reproducibility:**

4: Could mostly reproduce the results, but there may be some variation because of sample variance or minor variations in their interpretation of the protocol or method.

**Reviewer Confidence:**

4: Quite sure. I tried to check the important points carefully. It's unlikely, though conceivable, that I missed something that should affect my ratings.

**Typos Grammar Style And Presentation Improvements:**

Line 025 devote to ->are devoted to
Line 026 belong -> belonging
Line 069 pre-treained ->pre-trained
Line 170 take -> taking
Line 243 a slot type -> a slot type sequence, based on the context, it should be a sequence of slot types?
Line 269 in training batch -> in the training batch

Too many long sentences, for example line170-line 175. It would be better to use less many clauses in one sentences. The paper needs more polishing.

---

> ### Author Rebuttal · Authors · 2023-08-23
>
> Thanks for your review.
> Question A: A good question. We are the first case in your example:  ‘this week’ is labeled as date_time, date_time.
> Assume that coarse-grained slot type set is {'O': 0, 'date_time': 1, 'location': 2 } and fine-grained slot type set is {'O': 0, 'B-date_time': 1, 'I-date_time': 2, 'B-location': 3,  'I-location': 4 }. In coarse-grained loss,  both " this" and "week" share the same slot type *date_time* (we can use the digital number of dict value to represent it, i.e., so the coarse slot labels of "this week" is 1 1 ),  we directly use dict values of coarse slot type set (e.g., 0,1, or 2 ) as the training labels of coarse contrastive loss without encoding them, because these values can be directly used as supervised signals.  In this way, the proposed model will coarsely learn that which tokens are entities and which are not, and the difference between entities via supervised contrastive learning of entity-level label.
>
> Question B: Apologize for the confusion in Figure 3.  We divide your question B into 4 sub-questions, we respond them with a, b, c and d, respectively.
> &emsp;&emsp;a.   Each input instance (or sample) of transformer encoder consists of two parts, one slot type and one utterance. Since there are multiple slot types in one domain,  the volume of instances could be augmented via the combination of single utterance and multiple slot types. For instance, the slot type set of one domain is {'date_time', 'to_do', 'location'} , given a utterance "what the weather in st paul this weekend", three instances can be constructed: "data time [sep] what the weather in st paul this weekend",  "to do [sep] what the weather in st paul this weekend", and "location [sep] what the weather in st paul this weekend",   their BIO labels are "O, O, O, O, O, O , O, O, O, O, B, I", "O, O, O, O, O, O, O, O, O, O, O, O", "O, O, O, O, O, O, O, O, B, I, O, O",  respectively. Entity-level slot labels of three instances are "O, O, O, O, O, O, O, O, location, location, date_time, date_time" (three labels are the same) and their token-level slot labels  are "O, O, O, O, O, O, O, O, B-location, I-location, B-date_time, I-date_time" (three labels are the same) , in training the instances are randomly shuffled and allocated into different batch.
> For simplicity, we use two instances/utterances to represent a training batch ( the training batch size is two) to illustrate how entity-level or token-level contrastive learning works.
>
> &emsp;&emsp;b.  As described in a., each instance only contains one slot type. In CRF loss,  this slot type is related with tokens or entities of the utterance of the same instance ( this slot type has no relation with other instances) via BIO label, i.e., ["O", "B", "I"].  However,  in hierarchical (entity and token level) contrastive loss, we labeled this slot type as "O", therefore, it has no relation with other tokens or entities of both this instance and other instances except for "O"(outside) class.
>
> &emsp;&emsp;c. Slot types are labeled as outside or 'O' during entity-level and token-level contrastive learning, so the representations of slot types  transformed by Gaussian transformation network do not involve in token-class or entity-class training except for 'O' class.  However, the representations of slot type are trained during CRF loss, which help HiCL modeling the semantic relationship between slot_type and utterance via BIO three-tag label, i.e., ["O", "B", "I"].
>
> &emsp;&emsp;d. The slot labels of entity-level and token-level only serve for contrastive learning that based on Gaussian embedding, and the entity-level and token-level contrastive learning only perform during training ( not in inference), and they only calculate contrastive loss and predict nothing. While, in the prediction output of CRF,  the entity label is neither entity-level nor token-level, it is just a  BIO three-tag set ,  i.e., y_set = ["O", "B", "I"]  or  [0, 1, 2].  Here is an example,  given <slot type (e.g., date_time)+utterance (e.g., send a reminder ...)> as input, the HiCL outputs the location of this slot type in the utterance with the three-tag set. Specifically, given "date time [sep] send a reminder for a tire check next week" as model input, HiCL outputs (CRF in Figure 3) the label sequence of  "0, 0, 0, 0, 0, 0, 0, 0, 0, 0, 1, 2".
>
> Question C: In each dataset, we traverse all domains and make each of them as target domain,  i.e.,  each time make one domain as target domain and the rest domains are training domains, therefore target domain distributes across  all the domains. If you mean each time multiple domains as target domains, this setting never appeared in existing works of zero-shot slot filling of task-oriented dialogue, it is interesting but much more challenging. Especially for unseen slots, you know even only one target domain, the unseen-slots performance of  all the baselines is poor. Nevertheless, multiple domains as target domains are indeed an important indicator of cross-domain generalizability, which should be taken into consideration and continuously optimized in future works.
>
> Question D: If two tokens belong to one utterance, they are called intra-token towards each other.  Whereas, if two tokens belongs to two difference utterances, they are called inter-token towards each other. We carry the entity-level contrastive learning and entity-level contrastive learning in batch level during training, each batch may contains from several to dozens of utterances that depends on batch size setting.
>
> As to reasons to reject, we give the following supplements and explanations:
> We verified and summarized two slot-agnostic abilities for our proposed HiCL, discovering new entities and differentiating  between new token- or entity-class, respectively.
> We present a multi-dimensional and progressive analysis, to verify slot-agnostic ability step by step,  particularly, in Appendix M (Iterative Prediction Case Study),  we empirically and  qualitatively validate HiCL can differentiate  between new token- or entity-class（line 1349-line1358, which is a important ability of slot-agnostic ; again,  our study (line 1359-line1376) validates HiCL own more stronger capability of discovering new entities over baseline, which is also a critical ability of  slot-agnostic.
> In summary, slot-agnostic ability is a cross-domain generalization ability that can NOT be obtained by learning from slot-specific features ( all existing works just focus on slot specific learning and they almost do not have this ability). While, our proposed hierarchical contrastive leaning targets at learning powerful cross-domain slot generalization capacity and is specifically designed to learn self-agnostic ability for unseen slots.

---

### Meta-Review · Area_Chair_CJcC · 2023-09-19

**Recommendation:** 3

**Metareview:**

The paper presents a novel hierarchical contrastive learning approach for the zero-shot slot filling task. It introduces token-level contrastive learning and uses Gaussian embedding to improve generalizability in unseen domains. Reviewers appreciates the comprehensive nature of the paper and finds the idea of token-level contrastive learning innovative. The detailed ablation studies make the work convincing. However, they have some questions about the method and suggest more clarification and analysis. Reviewers acknowledges the solid experiments and good performance achieved. However, they point out a lack of MRC-based SF baselines, unclear descriptions, and a missing citation. They also have questions about the loss computation and entity-level representations.

Based on these reviews, the paper is considered of good soundness and shows some originality. However, there are concerns regarding clarity and missing references. The reviewers find the results and ablation studies compelling. The pros of the paper include the clear introduction, innovative token-level contrastive learning, and comprehensive ablation studies. The cons include the lack of clarification in the method section, missing citation for the contrastive loss, and unclear descriptions. Overall, the paper has potential but needs some revisions and clarifications.

---

### Decision · Program_Chairs · 2023-10-07

**Decision:**

Accept-Findings

**Comment:**

The paper presents a novel hierarchical contrastive learning approach for the zero-shot slot filling task. It introduces token-level contrastive learning and uses Gaussian embedding to improve generalizability in unseen domains. Reviewers appreciates the comprehensive nature of the paper and finds the idea of token-level contrastive learning innovative. The detailed ablation studies make the work convincing. However, they have some questions about the method and suggest more clarification and analysis. Reviewers acknowledges the solid experiments and good performance achieved. However, they point out a lack of MRC-based SF baselines, unclear descriptions, and a missing citation. They also have questions about the loss computation and entity-level representations.

Based on these reviews, the paper is considered of good soundness and shows some originality. However, there are concerns regarding clarity and missing references. The reviewers find the results and ablation studies compelling. The pros of the paper include the clear introduction, innovative token-level contrastive learning, and comprehensive ablation studies. The cons include the lack of clarification in the method section, missing citation for the contrastive loss, and unclear descriptions. Overall, the paper has potential but needs some revisions and clarifications.